# LEARNING LLM-AS-A-JUDGE FOR PREFERENCE ALIGNMENT

**Ziyi Ye[1], Xiangsheng Li[2], Qiuchi Li[3], Qingyao Ai[1,\*], Yujia Zhou[1], Wei Shen[2],**
**Dong Yan[2,\*], Yiqun Liu[1]**
[1]Department of Computer Science and Technology, Tsinghua University
[2]Baichuan AI   [3]University of Copenhagen
ye-zy20@mails.tsinghua.edu.cn, lixsh6@gmail.com, qiuchi.li@di.ku.dk, aiqy@tsinghua.edu.cn,
zhouyujia@mail.tsinghua.edu.cn, shenwei0917@126.com, weyshioncn@gmail.com,
yiqunliu@tsinghua.edu.cn

## ABSTRACT

Learning from preference feedback is a common practice for aligning large language models (LLMs) with human value. Conventionally, preference data is learned and encoded into a scalar reward model that connects a value head with an LLM to produce a scalar score as preference. However, scalar models lack interpretability and are known to be susceptible to biases in datasets. This paper investigates leveraging LLM itself to learn from such preference data and serve as a judge to address both limitations in one shot. Specifically, we prompt the pre-trained LLM to generate initial judgment pairs with contrastive preference in natural language form. The self-generated contrastive judgment pairs are used to train the LLM-as-a-Judge with Direct Preference Optimization (DPO) and incentivize its reasoning capability as a judge. This proposal of learning the LLM-as-a-Judge using self-generated **Con**trastive judgments (Con-J) ensures natural interpretability through the generated rationales supporting the judgments, and demonstrates higher robustness against bias compared to scalar models. Experimental results show that Con-J outperforms the scalar reward model trained on the same collection of preference data, and outperforms a series of open-source and closed-source generative LLMs. We open-source the training process and model weights of Con-J at https://github.com/YeZiyi1998/Con-J.

## 1 INTRODUCTION

As Artificial Intelligence (AI) systems based on Large Language Models (LLMs) are increasingly used in various applications, it is crucial to ensure they align with human instructions, values, and ethics. LLMs alignment is generally achieved by learning from preference data that compares pairs of responses to a question (Rafailov et al., 2024; Christiano et al., 2017; Liu et al., 2020). However, collecting high-quality human preference data is both time-consuming and costly. In practice, the construction of preference datasets often involves a combination of human and AI-generated feedback (Lee et al., 2023; Hou et al., 2024). Therefore, it is crucial to develop an efficient and accurate AI model for preference prediction that aligns with human values.

To obtain such preferences, industrial practices have used scalar models (Hou et al., 2024) that concatenate the pre-trained LLM with a value head to produce scalar scores as preference. However, the scalar model suffers from limitations, particularly in the following aspects: (i) *Lack of interpretability*: The scalar model does not provide any support or explanation for its judgment. This hinders human involvement for its evaluation and verification. (ii) *Susceptibility to bias*: It is prone to capturing the biases present in the preference dataset rather than human values. For example, when the majority of positive answers in preference datasets are longer sentences, the learned LLM will likely favor more verbose answers (Huang et al., 2024b).

---

This work is supported by Quan Cheng Laboratory (Grant No. QCLZD202301).
*Corresponding author.

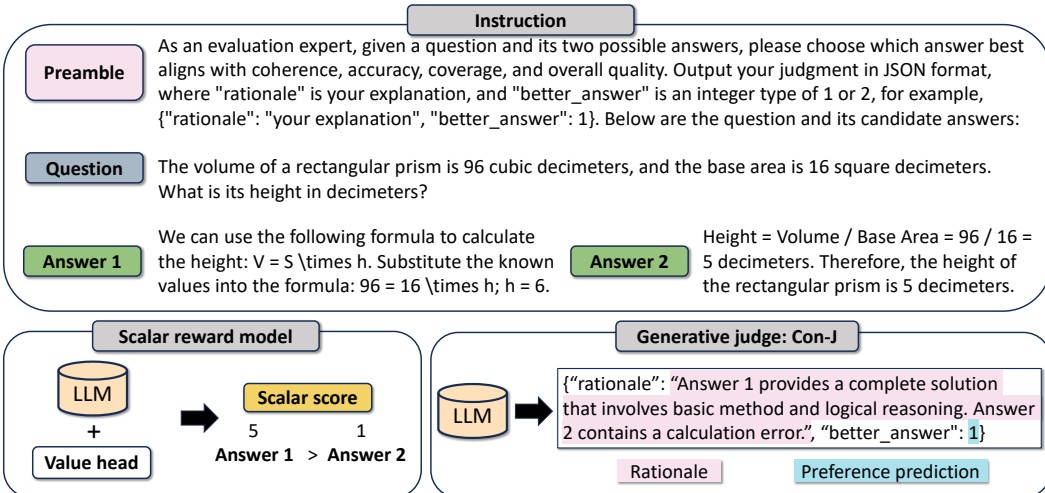

Figure 1: Top: Examples of a preamble, a question, a pair of answers, and the corresponding judgment (see the detailed version in Table 5). Bottom: Illustrations of a scalar reward model and the proposed Con-J for preference judgment.

To address the above limitations, we propose **Con-J**, which learns an LLM-as-a-**J**udge using its self-generated **Con**trastive judgments (see Figure 1). Con-J leverages the LLM's pre-existing judgment ability and bootstraps this ability for more accurate preference prediction. As shown in Figure 2, Con-J consists of three steps: (**Judgment Sampling**) Sample several judgments from a pre-trained LLM by prompting it with a query and a pair of candidate answers. (**Judgment Filtering**) Leverage the true preference annotations to construct contrastive judgment pairs, i.e., judgments with correct or incorrect preference. (**Contrastive Training**) Train Con-J from the pre-trained LLM based on these contrastive judgments using Direct Preference Optimization (DPO).

The design of Con-J differs from existing methods for learning the LLM-as-a-judge (or generative judge) Zhang et al. (2024); Kim et al. (2024); Park et al. (2024). These methods typically depend on external models (particularly GPT-4) or algorithmic schemes to produce high-quality instruction-tuning datasets. In contrast, Con-J directly learns from preference data using a self-bootstrapping approach without supervised fine-tuning on human instructions. As LLMs become more powerful, aligning them with high-quality instruction data becomes more difficult since humans may not always be able to write superior instructions. Instead, Con-J offers a new way by eliciting what the LLM already knows, supervised by human preferences, which are much easier to obtain than high-quality human instructions.

We train and evaluate Con-J on self-built commercial datasets across three domains: Creation, Math, and Code. We observe that Con-J not only outperforms the scalar model but also significantly surpasses GPT-4o across three domains. At the same time, Con-J can generate rationales to support its preference prediction. We conduct human and machine evaluations on the rationales in terms of their ***correctness***, i.e., whether the rationale provides an accurate analysis supporting the preference prediction, and ***consistency***, whether the rationale expresses the same preference as Con-J's final preference prediction. Experimental results show that preference learning improves the correctness of Con-J's rationales. Additionally, we found that Con-J is less susceptible to dataset biases, which we attribute to the regularizing effect inherent in Con-J's generative training target. To facilitate further research and development within the community, we train and release a public version of Con-J on publicly available datasets, which outperforms a series of existing open-source LLMs.

To summarize, our contributions are: (1) We propose Con-J, an approach that trains an LLM-as-a-judge using a self-bootstrapped technology to learn from preference data. (2) We show that Con-J can offer not only more accurate preference prediction but also more accurate rationales, unraveling a self-evolution process during preference learning. We also provide theoretical motivation and empirical evidence showing that Con-J is more robust to dataset biases due to its generative training target. (3) We test the performance of Con-J in commercial datasets and publicly available benchmarks. Con-J outperforms the scalar models and a series of existing LLMs.

## 2 PRELIMINARY

### 2.1 TASK DEFINITION

Given a question or prompt $q$ and a pair of assistant responses $a^1$ and $a^2$, the task is to judge the preference between $a^1$ and $a^2$. To accomplish this, we train the model using an existing preference dataset $D = \{(q, a^-, a^+)_i\}_{i=1}^N$, where $a^+$ is a preferred answer compared to $a^-$. The model's performance is subsequently evaluated on a separate, non-overlapping preference dataset by measuring the accuracy of its preference judgments.

### 2.2 SCALAR MODEL

The most common practice for getting the preference judgment is to use a scalar model (SM) similar to the reward model in the RLHF stage (Hou et al. (2024)). The SM predicts numerical scores $r(q, a)$ for $a \in \{a^1, a^2\}$ and judges the preference by comparing $r(q, a^1)$ and $r(q, a^2)$. It is typically initialized by concatenating a pre-trained LLM with a randomly initialized shallow MLP head. The most widely used training objective for the SM follows the Bradley-Terry model, which maximizes the probability of $a^+$ being preferred:

$$P(a^+ \succ a^- | q) = \frac{\exp(r(q, a^+))}{\exp(r(q, a^+)) + \exp(r(q, a^-))} = \sigma\big(r(q, a^+) - r(q, a^-)\big) \tag{1}$$

where $\sigma$ is the sigmoid function.

The above-mentioned SM utilizes the prompt $q$ and a single answer $a$ as input, which we denote as a pointwise SM. In addition, existing research has investigated a pairwise variant that uses a pair of candidates as input, i.e., $r(q, a^1, a^2)$ (Jiang et al., 2023). The pairwise vanilla reward formalizes the preference probability of $a^+$ as:

$$P(a^+ \succ a^- | q) = \sigma(r(q, a^+, a^-)). \tag{2}$$

To train the above-mentioned pointwise and pairwise SM $r$, previous studies maximize the log-likelihood of the preferences by minimizing the following loss function:

$$\ell_R(r) = - \sum_{(x, a^+, a^-)} \log p_r(a^+ \succ a^- \mid x) = \begin{cases} -\sum_{(x, a^+, a^-)} \log \sigma(r(x, a^+) - r(x, a^-)) & \text{(pointwise)} \\ -\sum_{(x, a^+, a^-)} \log \sigma(r(x, a^+, a^-)) & \text{(pairwise)} \end{cases} \tag{3}$$

### 2.3 LLM-AS-A-JUDGE

Instead of using a scalar model for preference judgment, existing literature also leverages the LLM itself as a generative judge to make preference judgments (Guo et al. (2024); Zheng et al. (2023); Li et al. (2024b); Gu et al. (2024); Li et al. (2024a)). Given the question $q$ and a pair of answers $a^1$ and $a^2$, a prompt $p$ is constructed by concatenating a preamble with $q$, $a^1$, and $a^2$. The preamble is an instruction that describes the task and asks an LLM $\pi$ to act as a judge (see examples in Appendix 5). Then the LLM generates natural language judgments $j = \pi(p)$. Next, regular expression matching is adopted to extract the preference prediction of the judgment $j$.

Some of the existing LLMs-as-Judges will generate supporting rationales before making the preference prediction (Li et al., 2023), while others directly output judgments according to a template (Shiwen et al., 2024). In our experiment, we construct the preamble by modifying the prompt from the existing research (Lee et al., 2023). The preamble requires the LLMs to output the judgment in a JSON style: a key named `"rationale"` includes step-by-step reasoning and explanation, and another key named `"better_answer"` indicates the LLM's binary preference.

## 3 IMPROVING LLM-AS-A-JUDGE BY TRAINING ON CONTRASTIVE JUDGMENTS

As shown in Figure 2, the construction of Con-J consists of three steps: *judgment sampling*, *judgment filtering*, and *contrastive training*.

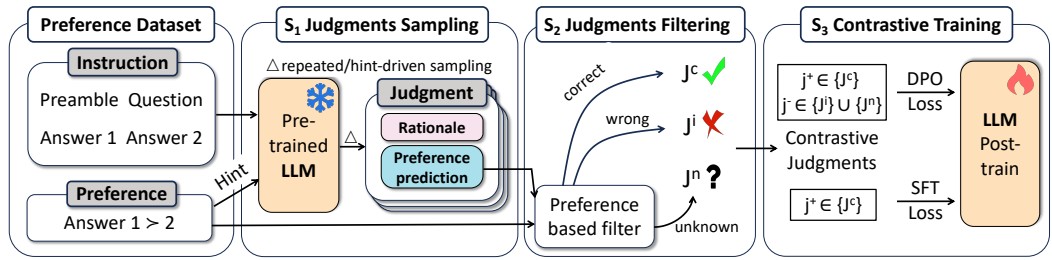

Figure 2: The steps for constructing Con-J with a preference dataset that includes preference annotations for a pair of answers to a question. $S_1$: Prompt Con-J to generate multiple judgments for a pair of answers by repeated sampling and hint-driven sampling. A judgment is composed of a preference prediction and a supporting rationale (see Figure 1 for an example). $S_2$: Bootstrap contrastive judgment pairs by filtering with true preference labels. $S_3$: Train Con-J using the DPO loss on contrastive judgments and the SFT loss on positive judgments.

($S_1$) *Judgment sampling:* We construct contrastive judgment pairs by prompting the LLM to generate multiple judgments. As shown in Figure 2, this is achieved by (1) *repeated sampling* and (2) *hint-driven sampling*. Repeated sampling prompts the LLM to generate multiple outputs from the same prompt, each utilizing a different random seed. However, the LLM may produce only one-sided judgments (i.e., all judgments preferred $a^1$ or $a^2$) across all repeated samples, making it impossible to construct contrastive judgment pairs. Therefore, we propose hint-driven sampling to compel the LLM to generate judgments that prefer specific answers. Essentially, the LLM is provided with an explicit indication of which answer is better, and is instructed to generate the judgment accordingly in JSON format as above. The prompt template for hint-driven sampling is provided in Table 6. By manipulating the hint, we can get a contrastive judgment pair for any prompt input.

($S_2$) *Judgment filtering:* We denote the outputs from repeating sampling as $M(p)$. $M(p)$ can potentially include both "positive" and "negative" judgments. A positive judgment indicates the preference prediction corresponding to the keyword `"better_answer"` is correct ($j^+$), while a negative judgment ($j^-$) indicates the preference prediction is incorrect ($j^i$) or the model doesn't explicitly indicate its preference ($j^n$). Contrastive judgment pairs $\{(j^+, j^-)\}$ are then constructed as the direct product of the positive judgment set $M(p)^+$ and negative judgment set $M(p)^-$. We set the number of repeated samplings to 8, allowing for the construction of up to 4 contrastive pairs (in the optimal case, there exist 4 positive and 4 negative judgments among the 8). For hint-driven sampling, we prompt the LLM with one correct and one incorrect hint and construct one pair from its judgments. The detailed sampling and filtering process is outlined in Algorithm 1.

($S_3$) *Contrastive training:* Based on the contrastive judgment pairs $D^J = \{(q, a^+, a^-, j^+, j^-)_i\}_{i=1}^{K}$, we train the LLM $\pi$ with a direct preference optimization (DPO) loss function:

$$\ell^{\text{DPO}} = -\sum_{(p, j^+, j^-)} \log \sigma \left[ \eta \log \frac{\pi(j^+|p)}{\pi_0(j^+|p)} - \eta \log \frac{\pi(j^-|p)}{\pi_0(j^-|p)} \right] \tag{4}$$

where $\pi_0$ is the reference model initialized as the base LLM and remains untrained. Following existing practice (Hong et al., 2024; Pal et al., 2024; Yang et al., 2024b; Huang et al., 2024a; Cen et al., 2024), the DPO also fuses a small weight of supervised fine-tuning (SFT) loss to help mitigate the overoptimization issue (Liu et al., 2024; Fisch et al., 2024), which can be formulated as:

$$\ell^{\text{SFT}} = -\sum_{(p, j^+)} \log \pi(j^+|p) \tag{5}$$

Then we linearly combine the DPO loss and the SFT loss with a small weight $\alpha$:

$$\ell^{\text{final}} = \ell^{\text{DPO}} + \alpha * \ell^{\text{SFT}} \tag{6}$$

**DPO training enables LLM to better distinguish different answers.** Existing methods to improve the judgment accuracy of an LLM are generally based on supervised fine-tuning (SFT) (Kim et al., 2024; Zhang et al., 2024; Li et al., 2023) to imitate correct judgments. However, we empirically find that SFT only is insufficient (see Setion 4). Intuitively, LLMs should identify the more

important aspects of a judgment, rather than patterns that may appear in both positive and negative judgments (Park et al., 2024). For example, both "answer 1 has logical errors, so the better answer is 2" and "answer 2 has logical errors, so the better answer is 1" could be positive judgments for different prompt inputs, even though they are opposite in meanings. When LLM is trained with SFT loss, it may primarily imitate the common pattern that appears in both answers rather than developing the ability to make judgments based on the prompt. Existing evidence [1] even suggests that the likelihood of generating negative output might even surpass that of positive output during SFT training.

**Rationales bring robustness against bias.** The proposed Con-J can generate rationales in addition to the binary preference prediction. We suggest that training the model to generate rationales can impart a regularization effect and help avoid potential biases in the datasets. Here we provide a theoretical motivation for this effect. We decompose a judgment $j$ into $j_r$ and $j_y$, representing the rationale and the binary preference prediction, respectively. Adding the rationales as training targets can be formalized by introducing an intermediate variable $j_r$ influencing the conditional probability $P_\theta(j_y \mid p)$:

$$P_\theta(j_y \mid p) = \sum_{j_r} P_\theta(j_y \mid j_r, p) \, P_\theta(j_r \mid p) \tag{7}$$

By including rationales, the bias in preference data is distributed between $j_y$ and $j_r$, reducing its direct impact on $j_y$. We can formalize the loss function as:

$$\ell(\theta) = -\sum_{(p,j)\in D} \log P_\theta(j_y \mid j_r, p) - \sum_{(p,j)\in D} \log P_\theta(j_r \mid p) \tag{8}$$

The loss $P_\theta(j_r \mid p)$ encourages the model to find effective representations for predicting $j_r$, exerting a regularizing effect compared to directly predicting the preference $j_y$.

**LLM-as-a-Judge resists bias with LLM's prior learning from pretraining.** SM modifies the LLM's architecture with a classification head and uses a discriminative training target. On the contrary, Con-J uses an architecture consistent with the pertaining process and generative training objectives (Zhang et al. (2024)). We refer to Erhan et al. (2009) and assume the parameter inherited from the pre-trained LLM as adding an infinite penalty:

$$\ell(\theta) = \ell_{data}(\theta) + \frac{\lambda}{2}||\theta - \theta_0||^2 \tag{9}$$

where $\lambda$ is the regularization strength. We make an ideal hypothesis that there exists an optimal $\theta^*$ which fully encodes human values and consistently makes true judgments. Hence we assume that the parameters obtained during the pre-training phase are closer to $\theta^*$ than a random distribution. This analysis suggests Con-J gets a smaller penalty term for optimization towards $\theta^*$. In contrast, SM adopts a different training objective and introduces a randomly initialized head, making the regularization effect less significant (smaller $\gamma$ and partially randomly initialized $\theta_0$). Hence, SM can be good at encoding the knowledge reflected in the preference dataset but is also more sensitive to its bias than Con-J.

## 4 EXPERIMENTS

### 4.1 EXPERIMENTAL SETUP

**Datasets.** We train scalar models and Con-J on three datasets within different vertical domains: Creation, and Math, then evaluate their performance in terms of the accuracy of preference predictions. The datasets are self-built commercial datasets consisting of approximately 120,000, 50,000, and 50,000 preference samples for Creation, Math, and Code, respectively. The Creative dataset involves tasks on text creation such as writing poetry or crafting headlines. The Math dataset mainly includes middle and high school math problem-solving tasks. The Code dataset comprises code-writing tasks in various programming languages and includes code-related problems. These datasets cover diverse sources, ranging from data generated by a commercial ChatBot, data generated by ChatGPT, and

---

[1]https://github.com/LLaMafia/SFT_function_learning

data from open-source datasets like HH-rlhf (Bai et al., 2022) and Infinity-Instruct[2]. The preference annotation for these datasets is gathered from human annotators, with each sample annotated by one annotator and subsequently verified by another. In addition to the self-built datasets, we train an open-source version of Con-J on a publicly available dataset Skywork-Reward-Preference-80K-v0.1[3] and test its performance on public benchmarks including Infinity-Preference[4], UltraFeedback (Cui et al., 2023) (select its test set according to HuggingFaceH4 [5]), PKU-SafeRLHF (Ji et al., 2024), and Reward-Bench (Lambert et al., 2024). We ensure that no identical prompts appear in both the training and test sets by filtering them out of the training set.

**Model.** We select Qwen2-7B-Instruct (Yang et al., 2024a) as the base model to train both the scalar model (named **SM**) and the proposed model (named **Con-J**). SM includes both pairwise and pointwise variants. Additionally, we included the original pre-trained Qwen2-7B-Instruct as an untrained variant of Con-J. We also compare Con-J with a range of LLMs-as-Judges, including GPT-4o[6] and two LLMs-as-Judges Auto-J (Li et al., 2023) and Prometheus 2 (Kim et al., 2024) trained by SFT, Llama series (Llama3.1-8B, and Llama3.1-70B), and Qwen series (Qwen2-7B, Qwen2.5-72B). The details about hyperparameters, sampling, and sampling strategy of Con-J is provided in Section A.1.

## 4.2 MAIN RESULTS

Table 1 presents the results of SM and Con-J on the self-built commercial datasets across three vertical domains. It can be observed that (i) there is no significant difference between the pointwise and pairwise variants of SM. Although existing research suggests that concatenating the list of responses improves performance for scoring the responses (Jiang et al., 2023), we do not observe this effect on our datasets. (ii) Both Con-J and SM outperform off-the-shelf GPT-4o, indicating that small models trained on domain-specific data can effectively predict domain-related preferences. (iii) On the same preference datasets, Con-J consistently outperforms SM across all tasks with a significant gap on the Text Creation task. This indicates that Con-J is more effective at acquiring accurate judgment abilities than SM.

Table 1: Judgment accuracy of GPT-4o, SM, and Con-J. $*$ indicates the performance difference between Con-J is significant at $p < 0.05$ using a pair-wise t-test.

| Model | Creation | Math | Code |
|---|---|---|---|
| GPT-4o | $55.6^*$ | $74.8^*$ | $68.1^*$ |
| SM (point-wise) | $69.4^*$ | 84.8 | 69.4 |
| SM (pair-wise) | $69.2^*$ | 84.6 | 69.6 |
| Con-J | **72.4** | **85.0** | **70.1** |

We carry out an ablation study to investigate the variants of Con-J. Given that current methodologies often employ Supervised Fine-Tuning (SFT) to train LLMs-as-Judges (Li et al., 2023; Kim et al., 2024), we developed an SFT variant of Con-J, trained exclusively on positive judgments using SFT loss. As illustrated in Table 2, Con-J trained with our proposed framework outperforms its variant without DPO loss across all datasets. This observation demonstrates the effectiveness of training from contrastive judgments. In addition, Con-J outperforms its variant without hint-driven sampling, which relies solely on repeated sampling and may be infeasible to construct contrastive judgment pairs for some

Table 2: Judgment accuracy of Con-J and its variants. $*$ indicates the performance difference between Con-J is significant at $p < 0.05$ using a pair-wise t-test.

| Model | Creation | Math | Code |
|---|---|---|---|
| Con-J untrained | $53.6^*$ | $63.4^*$ | $61.7^*$ |
| Con-J w/o Hint | $61.3^*$ | $77.4^*$ | 68.2 |
| Con-J w/o DPO | $54.6^*$ | $64.2^*$ | $63.5^*$ |
| Con-J | **72.4** | **85.0** | **70.1** |

prompts. Similar findings have been observed when using self-taught techniques to improve LLMs (Zelikman et al., 2022). More variants of Con-J were tested and detailed in Section A.2.

---

[2]https://huggingface.co/datasets/BAAI/Infinity-Instruct

[3]https://huggingface.co/datasets/Skywork/Skywork-Reward-Preference-80K-v0.1

[4]https://huggingface.co/datasets/BAAI/Infinity-Preference

[5]https://huggingface.co/datasets/HuggingFaceH4/ultrafeedback_binarized

[6]https://openai.com/index/hello-gpt-4o/

Table 3: Accuracy of generative judges on the test sets of four benchmarks: Infinity-Preference, UltraFeedback, PKU-SafeRLHF, and Reward-Bench. Results in **bold** are the best among all models and results with underline are the second-best.

| Model | Infinity-Preference | Ultra-Feedback | PKU-SafeRLHF | Reward-Bench | | | |
|---|---|---|---|---|---|---|---|
| | | | | Chat | Chat-H | Safety | Reasoning |
| Llama3.1-8B | 59.0 | 62.9 | 66.4 | 80.7 | 49.8 | 64.0 | 68.1 |
| Llama3.1-70B | 64.0 | 71.4 | 67.6 | **97.2** | 70.2 | 82.8 | 86.0 |
| Qwen2-7B | 59.0 | 64.5 | 67.2 | 91.3 | 44.8 | 73.6 | 69.0 |
| Qwen2.5-72B | 70.0 | 66.0 | 58.7 | 86.6 | 61.4 | 74.5 | **90.7** |
| Auto-J | 69.0 | 63.9 | 66.9 | 93.0 | 40.0 | 65.5 | 50.5 |
| Prometheus 2 | 68.0 | 63.3 | 63.0 | 85.5 | 49.1 | 77.1 | 76.5 |
| GPT-4o | 75.0 | 72.2 | **69.6** | 95.3 | 74.3 | 87.6 | 86.9 |
| Con-J (ours) | **81.0** | **73.0** | 68.4 | 91.3 | **79.6** | **88.0** | 87.1 |

For a fair comparison with other LLMs-as-Judges, we tested the open-source version of Con-J on public benchmarks, as shown in Table 3. Con-J outperforms existing open-source LLMs-as-Judges in the vast majority of benchmarks, including commercial instruction-tuned models such as Llama series, Qwen series, and a series of LLMs-as-Judges trained specially for preference judgments such as Auto-J and Prometheus 2, except in the Chat and Reasoning sub-task of Reward-Bench. Additionally, Con-J achieves comparable performance with the closed-source model GPT-4o across all benchmarks. Specifically, it surpasses GPT-4o in the Infinity-Preference and UltraFeedback benchmarks and also outperforms it in the average performance of the Reward-Bench.

## 4.3 PREFERENCE LEARNING YIELDS MEANINGFUL AND USEFUL RATIONALES.

We investigate the quality of rationales in the Math dataset because the evaluation of math problem-solving tasks is relatively more objective. We select 6 checkpoints trained on different numbers of contrastive judgment pairs, i.e., 2k, 4k, 8k, 16k, 32k, and 64k, with 64k being the final checkpoint. Then we prompt GPT-4o to score the rationale's correctness (1 to 5) and its consistency with the predicted preference (1 to 3) (see the prompt in Table 9). Additionally, GPT-4o is tasked as a meta-judge to judge whether Con-J makes correct preference predictions. If the judgments of GPT-4o conflict with the dataset's true preference annotations, we exclude these questions for further analysis, as these questions may exceed GPT-4o's capabilities.

Experimental results are presented in Figure 3. From Figure 3(a), we observe that **the correctness of the rationales improves when Con-J is trained with more data and achieves increased preference prediction accuracy**. However, we find that the consistency between Con-J's preference prediction and its rationales decreases with the increase in judgment accuracy, as shown in Figure 3(b). These observations indicate that Con-J's abilities to make binary preference predictions and generate correct rationales both improve with training. However, the increase in inconsistency indicates that these improvements may not be balanced. We suspect that supervision from preference datasets, focused solely on binary preference prediction, enhances Con-J's binary prediction ability more than its reasoning ability, leading to inconsistencies.

We further test whether the improved rationales can be used to help a weak judge make more accurate preference predictions. We use the untrained Con-J as the weak judge and prompt it with the rationales generated by the stronger model. As shown in Figure 3(c), the weak judge yields more accurate preference prediction with the rationales provided by a stronger model. Additionally, we find that rationales generated by GPT-4o can also improve the weak judge, with performance comparable to the rationales from the strongest checkpoint of Con-J. This indicates that Con-J not only surpasses GPT-4o in preference prediction performance but also generates rationales with similar effectiveness.

Motivated by the above observation and the fact that the training of Con-J is only supervised by preference labels, we further investigated whether controlling the quality of the rationales can help Con-J make better preference prediction, which is elaborated in Section A.4. **Experimental results indicate that by controlling the quality of Con-J's rationale, we can not only improve the quality of the rationale but also further enhance the accuracy of the preference prediction.**

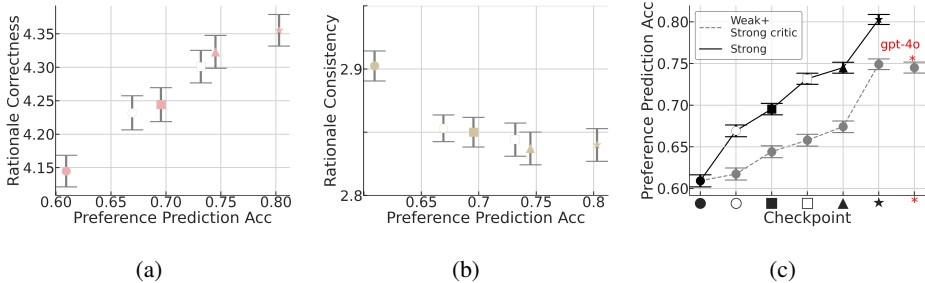

Figure 3: We analyzed 6 checkpoints (●, ◯, ■, □, ▲, ★) trained with different number of contrastive judgment pairs (2k,4k,8k,16k,32k,64k). (a-b) We prompt GPT-4o to evaluate the correctness of the rationales and their consistency with the predicted preferences. (c) We use the rationale from a strong model as input to help a weak model in its preference prediction.

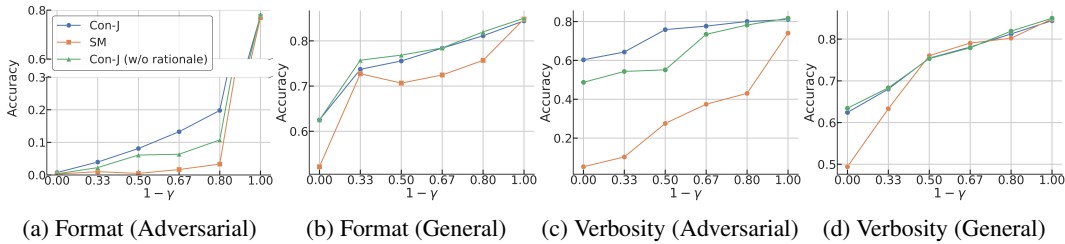

(a) Format (Adversarial)   (b) Format (General)   (c) Verbosity (Adversarial)   (d) Verbosity (General)

Figure 4: The performance of Con-J, Scalar model, and Con-J (without rationale), trained with varying degrees of training data bias in format and verbosity, where $1 - \gamma = 1$ indicates no bias. The performance was evaluated on two test sets: an adversarial set, which injected inversed bias treatments with the training data, and a general set, which is the original test set.

## 4.4   RATIONALES HELP CON-J BE MORE ROBUST TO DATASET BIAS.

Scalar models are known to be susceptible to biases in datasets, resulting in preference judgments that reflect biases in the pattern of data rather than true human values. To investigate the susceptibility of Con-J, we conduct a synthetic experiment that injects artificial bias into the data. We define the degree of bias in training set $\mathcal{D}_{\text{train}}$ as $\gamma$. This indicates that the $\gamma$ proportion of the data contains unexpected biases that don't necessarily reflect human value, while the rest $1 - \gamma$ of the data are randomly sampled from the original training set.

We consider two widely studied biases: format bias and verbosity bias (Park et al., 2024; Singhal et al., 2023; Shen et al., 2023) (see details in Section A.1). All experiments are conducted on a filtered Math dataset, where the filtering ensures that the solution in the preferred answer is correct, while incorrect in the less preferred answer. In this scenario, the correctness of the solution is prioritized over the format and verbosity of the answers. We investigate the performance of Con-J, SM, and Con-J trained and evaluated without (w/o) generating the rationales (see the prompt examples in Table 7) under the injections of the addressed bias. We trained them on a biased dataset and tested their performance on an adversarial test set $\mathcal{D}_{\text{test}}^{\text{adversarial}}$ and a general test set $\mathcal{D}_{\text{test}}^{\text{general}}$. $\mathcal{D}_{\text{test}}^{\text{adversarial}}$ is constructed by injecting a contrary bias to the test set. For format bias, the contrary bias means rewriting the preferred answer with a more formal format and the less preferred answer with a more verbal format. For verbosity bias, the less preferred answer is rewritten to be more verbose using the same procedure applied to the preferred answer in the training set. $\mathcal{D}_{\text{test}}^{\text{general}}$ is the original test set without any modification. All models achieved an accuracy of over 99% on the biased training dataset $\mathcal{D}_{\text{train}}$ when $\gamma = 1$, indicating the models are thoroughly trained and are strongly influenced by the bias.

We present the experimental results in Figure 4(a-d) with the following observations:

(i) Con-J outperforms SM when trained on biased data. Intuitively, Con-J does not train a value head from scratch; instead, it leverages the text generation capabilities of pre-trained LLMs. We further investigate this performance gain in the context of format bias (adversarial), with $\gamma = 0.33$. We

Table 4: Example of rationales and binary preference generated by Con-J. We prompt GPT-4o to annotate the correctness (ranging from 1 to 5) of the rationales and the consistency between the predicted preference and the rationale (ranging from 1 to 3). Refer to more examples in Table 11.

| | Prompt input | Rationale | Preferred |
|---|---|---|---|
| **Math (1)** | **Question:** What is the range of the numeric output of a sigmoid node in a neural network? 
 **Answer 1:** The output of a sigmoid node is bounded between -1 and 1. 
 **Answer 2:** The output of a sigmoid node is bounded between 0 and 1. | -1 in the first answer is incorrect because the output range of the sigmoid function is 0 to 1, not including -1. 
 *Correctness: 5, Consistency: 3* | Answer 1 (correct) |
| **Math (2)** | **Question:** Soldiers built a 2000-meter line. They averaged 350 meters per day for the first 3 days. What is the required daily average for the remaining 2 days? 
 **Answer1:** $(2000 - 350 \times 3) \div 2 = 1400 \div 2 = 700$ meters. Answer: An average of 700 meters needs to be set up each day for the last two days. 
 **Answer 2:** $(2000 - 350 \times 3) \div 2 = (2000 - 1050) \div 2 = 950 \div 2 = 475$ meters. Answer: 475 meters for the last two days. | The first answer is straightforward and accurate, taking into account the total distance that needs to be set up and the known information; the second answer offers a slightly simplified expression and the result is correct. 
 *Correctness: 2, Consistency: 2* | Answer 2 (correct) |

observe that when Con-J outperforms SM, the untrained model achieves 85.6% accuracy compared to 49.8% when Con-J underperforms SM. This suggests that Con-J is likely to generate rationales aligning with preferences from its pre-trained data. However, this does not mean that Con-J's ability to resist bias compared to SM is entirely due to the initial model. As shown in Figure 4(b), Con-J's performance is noticeably better than the initial model when $\gamma <= 0.67$. **This indicates that Con-J can not only retain knowledge from pretraining but also extract useful judgment standards even from biased data.** However, this phenomenon also raises potential challenges when we need to align LLMs to preferences with fundamentally different values or principles, which we left as future work.

(ii) When tested on the adversarial datasets, Con-J w/o rationales significantly underperforms Con-J when $\gamma > 0.2$ and $\gamma > 0.33$ for the format bias and the verbosity bias, respectively. **This indicates that Con-J becomes more robust at learning from biased data through training with rationales.**

(iii) When tested on general test sets, Con-J w/o rationales demonstrates comparable performance to Con-J with rationales. We conduct a further analysis under the verbosity bias with $\gamma$=1.0 which categorizes the data samples into groups where the chosen answer is longer or shorter than the rejected answer. We find that Con-J w/o rationale achieves an accuracy of 47.14% when the chosen answer is shorter than the rejected answer, whereas Con-J achieves an accuracy of 58.71% (see Table 12). **This implies that, although the average performance of Con-J and Con-J w/o rationale is comparable, Con-J w/o rationale is more significantly impacted by bias.**

(iv) Some existing research suggests that rationales or critics generated ahead of preference judgments can facilitate Chain-of-Thought (CoT) reasoning and help the LLM make better judgments (Lee et al., 2024; Ankner et al., 2024; Ye et al., 2024; Zhang et al., 2024). A possible explanation for the lack of a similar CoT effect in our dataset is that the CoT process is often already embedded in the responses, making the CoT procedure for judgment potentially unnecessary, which we further discussed in Section A.1.

## 4.5 HUMAN EVALUATION & CASE ANALYSIS

We further conduct a human evaluation on the Math dataset to investigate the quality of the rationales (see Section A.3 for details about annotator recruitment and analysis). The human evaluation involves two steps: (1) a quantitative annotation to assess the rationale's correctness and consistency using Likert scales, resembling the annotation performed by GPT-4o in Section 4.3, and (2) a qualitative analysis where annotators openly discuss the main issues that exist in the rationales. For the quantitative annotation, Krippendorff's $\alpha$ values between GPT-4o and human annotators are 0.4427 for correctness and 0.6495 for consistency, indicating moderate and substantial agreement, respectively. **This suggests a consistent trend between evaluations made by humans and GPT-4o, despite some remaining disagreements.** Such differences may arise from the limitations of GPT-4o. For example, we observe that the $\alpha$ for correctness annotation is pretty low (0.1301) in cases where GPT-4 fails to make correct preference predictions.

In the qualitative analysis, annotators agreed that most of the rationales were correct and showed high consistency with the preference prediction, indicating that these rationales can support Con-

J's preference prediction (e.g., Math (1) in Table 4). In addition to the high-quality rationales, we especially focused on the cases where the rationale was assigned a low consistency (consistency 1 or 2) or low correctness (correctness not higher than 4). The annotators deemed that in these cases, the rationales provided by Con-J may fail to identify key differences between the two answers, and therefore express no explicit attitude toward either answer (e.g., Math (2) in Table 4). Indeed, we observed that the average accuracy of the preference prediction for low consistency cases is 59.9%, which is significantly lower than the average performance (85.0%) but higher than chance. **This indicates that Con-J sometimes struggles to provide a high-quality rationale, even when it makes a correct preference prediction.**

## 5 RELATED WORK

**LLM alignment.** The initial approach developed for aligning LLMs with human values was reinforcement learning from human feedback (RLHF) (Christiano et al., 2017; Liu et al., 2020). This technique involves training a scalar reward model (RM) and then using reinforcement learning (RL) to optimize a policy according to the RM. Recently, a series of direct alignment from preference (DAP) works, such as DPO (Rafailov et al., 2024), SiLC (Zhao et al., 2023), and IPO Azar et al. (2024), have gained popularity. Unlike RLHF, DAP methods directly update the LLM using preference data, which is simpler and more stable. To scale the preference datasets, it is common to train an external machine model from existing preference datasets (Hou et al. (2024); Wu et al. (2024)). This online and scalable construction process enables DPO to be deployed in an iterative setting (Xiong et al. (2024); Xu et al. (2023)) or online setting (Guo et al. (2024)). Therefore, developing an accurate external model for preference judgment is a significant problem.

**LLM-as-a-judge.** Instead of training a scalar model for preference judgment, employing LLMs as a generative judge has been a promising alternative (Zheng et al., 2023; Ye et al., 2023). Efforts have been made to train language models specialized in evaluations. For example, Li et al. (2023); Kim et al. (2024) construct instruction-tuning datasets by prompting GPT-4 and using supervised fine-tuning (SFT) to train a pre-trained LLM as a generative judge. Mahan et al. (2024) utilizes a similar self-bootstrapping idea and investigates the feasibility of utilizing such generative judge into RLHF pipeline. Our contribution to the existing research is that Con-J uses self-sampled contrastive judgments under the supervision of preference data, with experiments showing its robustness and interoperability.

## 6 DISCUSSIONS AND CONCLUSIONS

We introduced Con-J, a novel approach that trains an LLM-as-a-Judge by self-bootstrapped learning from preference data. Con-J addresses the limitations of scalar reward models, including lack of interpretability and susceptibility to dataset bias. Our experiments on commercial datasets across Text Creation, Math, and Code domains, as well as publicly available benchmarks, demonstrate the effectiveness of Con-J. Moreover, we show that the correctness of the rationales generated by Con-J improves during learning from preference data. This enables Con-J not only to make accurate judgments but also to provide reasonable explanations, potentially facilitating human-in-the-loop supervision of LLM alignment. Finally, we found that Con-J is less susceptible to biases in datasets compared to its variants without rationales and the scalar models.

As AI systems become more powerful, many suggest that they will reach the point at which human are unable to easily and reliably assess the quality of their outputs (Casper et al., 2023). To address this issue, using another AI to supervise itself is a viable solution; however, researchers suggest that these methods may fail without human involvement (Shumailov et al., 2024). This paper contributes to addressing this issue in two ways. On the one hand, Con-J can be used to supervise LLMs by acting as a judge. At the same time, Con-J produces an explanation of its output that is legible to humans or another trusted system. This indicates that we can spot errors made by Con-J. On the other hand, the training and construction of Con-J rely solely on preference data, which is easier to acquire from human annotators than high-quality instruction tunning data. In many cases humans often find it difficult to provide verbal reasons for their preference, the training of Con-J could be integrated with human preference annotations, thereby reducing human effort for LLM alignment.

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

## A APPENDIX

### A.1 EXPERIMENTAL SETUP

**Hyper parameters.** We train SM and Con-J from with the DeepSpeed library Rasley et al. (2020), Zero Redundancy Optimizer (ZeRO) Stage 3 Rajbhandari et al. (2020), gradient-checkpointing Chen et al. (2016) and FlashAttention Dao et al. (2022). We use bfloat16 (BF16) and tfloat32 (TF32) mix computation precision. We use a cosine scheduler for the learning rate and 3% warmup. The maximum sequence length is set as 4,096. The batch sizes for SM and Con-J are set to 128 and 24, respectively, while their peak learning rates are set to $9 \times 10^{-6}$ and $5 \times 10^{-7}$, respectively, in accordance with existing practices for 7B models.

For Con-J, we linearly combine the SFT loss and the DPO loss with $\alpha = 1e^{-6}$. Here we select the weight of the SFT component according to the reward curve of the chosen response (i.e., positive judgment in our paper). We observed the changes in the chosen reward during training and selected the smallest alpha that would not cause significant changes in chosen reward. This selection results in a relatively small weight than existing work. In previous research of DPO training, the prompts, and chosen responses had diverse formats, whereas in our training, the chosen responses follow a fixed json-style template with a key "rationale" and a key "better_answer". So if the coefficient for the SFT component is too large, it can easily cause the chosen reward to increase rapidly, which could potentially damage the model's performance.

**Sampling and inference strategy for Con-J.** We use VLLM Kwon et al. (2023) for the inference for Con-J. During repeated sampling and hint-driven sampling, we employ greedy sampling with top-p set at 0.9 and top-k at 20, with a maximum output length set as 512, a temperature of 1.0, and a repetition penalty of 1.2. During the evaluation, we set top-p at 1.0 and a temperature of 0.0.

Note that as the maximum output length is set as 512. In our experiments, we observed that when the initial model needs to produce a lengthy judgment, it might get cut off, leading to an "unknown judgment". Rendering such judgment as negative judgment is actually beneficial, as when failing to do so, we empirically found that the Con-J's outputs will become increasingly lengthy during preference learning. However, simply setting a fixed output length for any input may affect performance on tasks requiring longer outputs to achieve a chain-of-thought (CoT) effect. Although our empirical findings indicate that rationales have an insignificant CoT effect on Con-J, future work could explore the output length of an LLM beyond the scenarios studied in our research.

**The design of bias injection** For format bias, we use two different prompts to rewrite the pair of answers (see prompt instructions in Table 8). Specifically, we rewrite the preferred answer with a more verbal format and the less preferred answer with a more formal format with GPT-4o. For verbosity bias, we follow existing practice (Zheng et al., 2023) by asking GPT-4o to rephrase the preferred answer without adding any new information and insert the rephrased text at the end of the original answer.

### A.2 VARIANTS OF CON-J

We test several variants of Con-J, the variants are listed as follows: (i) Con-J untrained: the original pre-trained Qwen2-7B-Instruct. (ii) Con-J w/o Hint: (iii) Con-J w/o DPO: an SFT variant of Con-J, trained exclusively on positive judgments using SFT loss. (iv) Con-J w/o SFT: a DPO variant of Con-J, trained exclusively with DPO loss, i.e., the combination parameter $\alpha = 0$. (v) Con-J w/o rationale: a variant using the prompt template in Table 7, requiring the LLM output only the binary judgment without any rationales. (vi) Con-J w/o reorder: average the performance with and without reordering answer 1 and answer 2 in the input prompt.

Experimental results are presented in Table 10. Con-J outperforms its variant without hint-driven sampling, demonstrating the importance of ensuring all preference data can be used to construct contrastive judgments. Con-J also outperforms its variant without DPO loss. This observation demonstrates the effectiveness of training from contrastive judgments.

However, Con-J shows comparable performance with Con-J w/o SFT in Creation and Code, while outperforms Con-J w/o SFT in Math. We further analyze the averaged reward of positive and negative judgments (chosen and rejected) in the test set. For Con-J without SFT, we observe that the

chosen rewards are relatively small in the Creation and Code datasets, with values of -0.73 and -0.46, respectively. However, in the Math dataset, the chosen rewards decrease to -2.37. The observation in the Math dataset aligns with previous research indicating that if SFT loss is helpful for stabilizing DPO training by preventing the decrease of the log probability of chosen response.

In addition, we find that the performance difference between Con-J and Con-J w/o rationale is not significant. This indicates that Con-J can effectively learn only the preference prediction without rationales. Although some existing research suggests that rationales might have a CoT effect to boost preference prediction performance, this is not the case with our dataset. A possible explanation for the lack of a similar CoT effect in our data is that the CoT process is often already embedded in the responses, making the CoT procedure for judgment potentially unnecessary.

Finally, we find that reversing the order of answers has no significant impact on our datasets. Actually, we performed a random shuffle on the pair of answers when constructing the prompts. We found that Con-J's preference predictions on the test set were roughly 1:1 for the first and the second answers.

### A.3    Human evaluation

We conduct a human evaluation to evaluate the rationales. The human evaluation consists of two steps: a quantitative analysis to evaluate the rationale's correctness and consistency with the final preference prediction, and a qualitative analysis where annotators openly discuss the annotation experience and the usefulness and main issues of the rationales, such as the underlying reasons for data samples with low consistency.

#### A.3.1    The quantitative analysis

In the quantitative annotation task, the annotators are required to annotate the correctness with a 5-point Likert scale and the consistency with a 3-point Likert scale, which follows the instructions for GPT-4o (see Table 9). To kick off this task, the author team initially annotated 10 judgment examples generated by Con-J and conducted a discussion to reach an agreement on the annotation criterion. Then, we recruited 6 annotators from the college students. All the annotators are graduate students (masters or PhD candidates) with mathematics skills at the level of university engineering students. Firstly, the annotators are required to read the instructions including the definitions of correctness and consistency (see Table 9). Then, they will review the annotations for the 10 judgment examples to ensure that they have understood the annotation criterion. Then, we construct 100 data samples randomly sampled from the test set of the Math dataset, which contains a question, a pair of answers, a judgment made by Con-J, and a judgment made by an untrained version of Con-J. Each annotator will annotate 50 out of the 100 data samples, ensuring that each sample is annotated by three different annotators. To avoid position bias, we randomized the order of the data samples and the sequence of judgments (belonging to Con-J or Con-J untrained). On average, each annotation took about 3 minutes and each participant spent about 2.5 hours to accomplish the annotation task. They were paid 40 dollars for their effort.

The average correctness score and consistency score evaluated by human annotators are 3.22 and 2.89, respectively. We calculated the inter-annotator agreement among human annotators and the agreement between the human majority vote and the machine annotations in terms of Krippendorff's $\alpha$. The $\alpha$ values for human annotators are 0.7942 for correctness and 0.8508 for consistency. These imply that there is strong agreement among human annotators on this annotation task. Further, we calculated the agreement between the annotations from GPT-4o and the human's majority voting (select the median value when the three annotations are all different). The $\alpha$ values are 0.4427 for correctness and 0.6495 for consistency, indicating moderate and substantial agreement, respectively. This implies that there is a consistent trend between human annotators and GPT-4o, although some disagreements remain.

We further analyzed these disagreements to investigate their potential causes. First, the quality of the correctness annotations depends on the ability of humans and machines to make correct preference predictions towards the pair of answers. Therefore, we categorized the data samples into two groups: those where GPT-4o, when prompted as the judge, made correct judgments, and those where it made incorrect judgments. We observed that in data samples where GPT-4o made correct judgments, its agreement with human annotators was higher than average ($\alpha$ =0.6201). However, in cases where

GPT-4 fails to make correct judgments, its agreement with human annotators is low ($\alpha$=0.1301). This observation suggests that we should not rely on GPT-4's annotation when the task may exceed its capabilities. Therefore, we excluded these samples from our experiments in Section 4.3. Second, We manually analyzed the disagreed consistency annotations made by humans and GPT-4o. We found in all of these data samples GPT-4o assigns a higher consistency score than the human annotator. For example, GPT-4o may overscore some rationales that do not specify which responses are being evaluated (e.g., "The answer provided a clear calculation process and gave an accurate answer, with more standard mathematical expressions and a clear logical order"). This means that GPT-4o sometimes treats the rationales as consistent even if they lack of direct supporting factor for the preference prediction. We believe this issue could be solved through techniques such as few-shot prompting or reflection, which we leave as future work.

Next, we investigate and compare the annotation scores for Con-J and Con-J untrained. We found that the average rationale score generated by Con-J (3.47) was significantly higher than that of Con-J untrained (2.95), while the consistency of the rationale generated by Con-J (2.87) was slightly lower than that of Con-J untrained (2.91). The results are consistent with those evaluated using GPT-4o, indicating that through preference learning, the rationales generated by Con-J improve correctness but may slightly reduce consistency. However, we also observed differences between the observations obtained from human evaluation and those from GPT-4o evaluation. For example, Con-J shows a greater improvement in correctness with preference learning under human evaluation compared to GPT-4o. This difference indicates that humans and GPT-4o may have some different criteria in their evaluations, which we further discuss in the following section.

### A.3.2 THE QUALITATIVE ANALYSIS

After the quantitative annotation, we further conduct a qualitative analysis in which annotators openly discuss the annotation experience and the usefulness and main issues of the rationales. 3 out of the 6 annotators who participated in the quantitative task agreed to join this discussion. In this process, annotators are presented with the annotations made by GPT-4o and other annotators. Firstly, the annotators discussed cases in which they disagreed on annotations. Most of the cases mainly arose from minor differences in the estimation of rationales. For example, there were cases where some participants gave a score of 4 (Mostly correct) while others gave a score of 5 (Completely correct) for the correctness of the rationales. This occurred because participants might have had differing understandings of whether the rationale addressed the most important aspects for making the preference prediction. Further, the annotators discussed cases in which they disagreed with GPT-4o's annotation. They generally agreed that these disagreements arise due to the mistake made by GPT-4o, particularly in overestimating rationales that have the correct attitude toward one of the answers but are based on incorrect reasoning, and in overestimating inconsistent responses as highly consistent.

Second, we analyzed the data samples with poor consistency (consistency scores 1 and 2). We find that in these cases, Con-J often chooses to refrain from clearly expressing its preference, e.g., refuses to indicate which answer to refer to: "The answer provided a clear calculation process and gave an accurate answer, with more standard mathematical expressions and a clear logical order.", and presents the weakness of both answers: "Answer 1 did not accurately understand the original mathematical problem; Answer 2 overly complicated the solution form, neglecting a more straightforward calculation method." Furthermore, we examined the accuracy of Con-J's final preference prediction in cases where consistency scores assigned by GPT-4o were 1 or 2. We found that the average accuracy for these cases is 59.9%, which is significantly lower than the average performance (85.0%). This suggests that low consistency generally occurs when Con-J lacks sufficient judgment ability, resulting in the generated rationale lacking an explicit attitude towards either answer and supporting the final preference prediction.

Finally, we analyzed cases where Con-J made correct preference predictions but the correctness of the rationales is not greater than 4. We found that for cases with a correctness score 4, the rationale usually has a correct attitude towards either answer but might lack some important aspects. Here we provide an example case Math (4) in Table 11 in which the rationale indicates the correctness of answer 1 but does not explicitly illustrate the mistake made by answer 2.

Additionally, for the vast majority of cases with a correctness score of 3 or fewer, we observe that the rationale does not clearly indicate an explicit attitude towards either answer. In this case, the con-

sistency of these cases is also relatively low, e.g., the case presented in Table 4 Math (2). Actually, we also observe a positive correlation (Pearson's $r$=0.34) between the correctness and consistency of the rationales when Con-J made an inaccurate preference prediction. This indicates that Con-J is struggling in making the preference prediction for the task, resulting in the generated rationale showing bad quality.

### A.4 CON-J IMPROVES WITH CONTROLLED RATIONALES

To test whether controlling the quality of the rationale can further improve Con-J's ability, we conduct a pilot experiment where the quality of the rationales with the annotation of consistency and correctness from GPT-4o (see Section 4.3). The control step is adopted on the data samples constructed from the judgment sampling process (step 3 in Figure 2). Considering the high cost of using GPT-4o for annotation, we randomly selected a subset from the constructed data samples in the Math task. The subset consists of 12,000 data samples in total. Each data sample consists of a question, a pair of answers, a positive judgment, and a negative judgment. We adopt GPT-4o to annotate the positive judgment and preserve samples in which GPT-4o agreed that the positive judgment is correct. This filtering step preserved a total of 9,050 samples. We then categorized these samples into two groups: a high-quality group with the rationales' correctness score of 5 and a consistency score of 3, and a low-quality group with the remaining data samples. This step constructs 5869 data samples for the high-quality group and 3181 for the low-quality group. We then randomly select and retain 3181 data samples from the high-quality group to ensure that both groups have the same size. Then we use the proposed contrastive training method to train Con-J with high-quality group (denoted as Con-J-h) and low-quality group (denoted as Con-J-l), respectively.

Experimental results show that the accuracy of preference prediction is 0.6759 for Con-J-h and 0.6427 for Con-J-l. This implies that improving the quality of the rationales can also help the binary preference prediction. We then test the quality of the rationales generated by Con-J-h and Con-J-l, respectively. We observe that the average correctness of the rationales generated by Con-J-h (4.25) is higher than Con-J-l (4.15) and Con-J untrained (4.145). This observation opens new avenues and benefits for human involvement in understanding preference models and enhances their ability not only to make accurate preference judgments but also to base those judgments on correct reasoning. However, we observe that the consistency score of Con-j-h (2.85) is not higher than Con-J-l (2.87) and Con-J untrained (2.90). According to the human evaluation, Con-J may avoid expressing clear preferences in the rationale part when it lacks the ability to make judgments. Actually, the low consistency may not necessarily be bad, as it potentially indicates that the model is not convincing in its judgment and may need more human involvement for these cases. In future work, a more effective data construction process may address this problem by designing more appropriate expected rationales for these situations, such as having the model output "I find it difficult to judge, but I can provide some analysis ...".

## B LIMITATIONS

Several limitations of this work guide future directions including: (i) We demonstrate that preference learning can enhance the model's ability to generate correct rationales. Another unresolved and intriguing question is whether enhancing the quality of rationales could also improve the model's preference prediction abilities. It is an important problem to enhance the model's ability not only to make accurate preference predictions but also to base those judgments on correct reasoning. (ii) We demonstrated that Con-J can more effectively resist bias than SM in an adversarial experiment. However, further analysis is needed to understand why Con-J outperforms SM on complex, realistic datasets, and whether this is also related to bias. (iii) We suggest that Con-j can potentially facilitate human collaboration through interpretable preference judgments for LLM training. The design of such a pipeline is another interesting and valuable direction.

## C ETHICS STATEMENT

With the advancement of LLM capabilities and alignment techniques, it is important to discuss the ethics issues in this paper. Here are some potential issues to consider:

1. *LLM Bias*: LLM is prone to be biased, which raises significant ethical concerns that demand careful consideration to ensure its responsible usage. In addition to training methods and model architecture, the origin of bias is significantly related to the training data. The societal biases in the training data may affect the functioning of LLMs, resulting in outputs that accentuate stereotypes or involve unjust discrimination against certain groups. This paper investigates the bias in preference learning in a synthetic dataset and makes one hypothesis that the post-training preference dataset may contain serious bias. However, it is important to note that biases, particularly those related to majority and minority groups, are also prevalent during the pre-training. Therefore, we suspect that future work requires more rigorous dataset construction to reduce potential bias and feedback mechanisms to detect and address bias.

2. *AI Transparency*: Nowadays AI is widely applied in decision-making, planning, and various related tasks. Ensuring the transparency and trustworthiness of AI in this process is important. The proposed Con-J enhances the interpretability of AI-based preference annotation by generating natural language rationales to support its preference prediction. This transparency allows humans to understand the reasoning behind AI decisions. However, we have also observed that the consistency between the Con-J's judgments and its rationales decreases with preference learning. This poses a challenge to building transparent AI systems.

3. *LLM misinformation:* Large language models (LLMs) are prone to generating misinformation without a factual basis. This misinformation may also be low in checkability, making it difficult for humans to verify (Hendrik Kirchner et al., 2024). Therefore, we have a significant concern that the rationales supporting the LLM's preference predictions might be faulty and difficult to detect. Future efforts to analyze the rationales from the LLMs, and devising methods incorporating a fact-checking process to help mitigate the risk of spreading misinformation are important.

## D    REPRODUCIBILITY

Open-source model and code are available at: https://github.com/YeZiyi1998/Con-J. The code used in this paper is available under the Apache 2.0 license. All the experiments in this paper are carried out based on open-source frameworks, including Open-RLHF (Hu et al., 2024), Pytorch, and Transformers [7].

---

[7]https://huggingface.co/docs/transformers/index

Table 5: An example of a prompt fed to the LLM to generate preference reward judgments, consisting of a preamble (introduction and instructions describing the task), a question, and a pair of candidate answers. The preamble is neutral and does not explicitly indicate which answer is better.

| | |
|---|---|
| Preamble | As an evaluation expert, given a question and its two possible answers, please choose which answer best aligns with coherence, accuracy, coverage, and overall quality. Output your judgment in JSON format, where "rationale" is your explanation, and "better_answer" is an integer type of 1 or 2, for example, {"rationale": "your explanation", "better_answer": 1}. Below are the question and its candidate answers: |
| Question | The volume of a rectangular prism is 96 cubic decimeters, and the base area is 16 square decimeters. What is its height in decimeters? |
| Answer 1 | We can use the following formula to calculate the height h of the rectangular prism: V = S \times h. Substitute the known values into the formula: 96 = 16 \times h; h = 6. Therefore, the height of the rectangular prism is 6 decimeters. |
| Answer 2 | 5. Height = Volume / Base Area = 96 / 16 = 5 decimeters. |
| Prompt | *As an evaluation expert, given a question and its two possible answers, please choose which answer best aligns with coherence, accuracy, coverage, and overall quality. Output your judgment in JSON format, where "rationale" is your explanation, and "better_answer" is an integer type of 1 or 2, for example, {"rationale": "your explanation", "better_answer": 1}. Below are the question and its candidate answers:* 
 ***Question:*** *The volume of a rectangular prism is 96 cubic decimeters, and the base area is 16 square decimeters. What is its height in decimeters?* 
 ***Answer 1:*** *We can use the following formula to calculate the height h of the rectangular prism: V = S \times h. Substitute the known values into the formula: 96 = 16 \times h; h = 6. Therefore, the height of the rectangular prism is 6 decimeters.* 
 ***Answer 2:*** *5. Height = Volume / Base Area = 96 / 16 = 5 decimeters.* |

Table 6: Prompt template with preamble using correct or incorrect hints, where $\alpha, \beta$ are the ID of correct and incorrect answers, respectively, $\{\alpha, \beta\} = \{1, 2\}$, {{Question}}, {{Answer 1}}, {{Answer 2}} are the text content of the question, answer 1, and answer 2, respectively. When the LLM does not output a valid JSON format as expected (e.g., "rationale": "your explanation", "better_answer": $\alpha$), we use an alternative prompt (rows 3-4) to prompt it again and insert its output as the rationale into the template.

| | |
|---|---|
| Prompt with preamble_correct | *As an evaluation expert, given a question and its two possible answers, please choose which answer best aligns with coherence, accuracy, coverage, and overall quality. Below are the question and its candidate answers:* **Question:** {{Question}} **Answer 1:** {{Answer 1}} **Answer 2:** {{Answer 2}} *Given that answer $\alpha$ is better than answer $\beta$, please provide the rationale. Output your judgment in JSON format, where "rationale" is your explanation, and "better_answer" is an integer type of $\alpha$, for example, {"rationale": "your explanation", "better_answer": $\alpha$}.* |
| Prompt with preamble_incorrect | *As an evaluation expert, given a question and its two possible answers, please choose which answer best aligns with coherence, accuracy, coverage, and overall quality. Below are the question and its candidate answers:* **Question:** {{Question}} **Answer 1:** {{Answer 1}} **Answer 2:** {{Answer 2}} *Given that answer $\beta$ is better than answer $\alpha$, please provide the rationale. Output your judgment in JSON format, where "rationale" is your explanation, and "better_answer" is an integer type of $\beta$, for example, {"rationale": "your explanation", "better_answer": $\beta$}.* |
| Prompt with preamble_correct (alternative) | *As an evaluation expert, given a question and its two possible answers, compare the answers according to their coherence, accuracy, coverage, and overall quality. Below are the question and its candidate answers:* **Question:** {{Question}} **Answer 1:** {{Answer 1}} **Answer 2:** {{Answer 2}} *Given that answer $\alpha$ is better than answer $\beta$, please provide the rationale:* |
| Prompt with preamble_incorrect (alternative) | *As an evaluation expert, given a question and its two possible answers, compare the answers according to their coherence, accuracy, coverage, and overall quality. Below are the question and its candidate answers:* **Question:** {{Question}} **Answer 1:** {{Answer 1}} **Answer 2:** {{Answer 2}} *Given that answer $\beta$ is better than answer $\alpha$, please provide the rationale:* |

Table 7: Prompt template for asking the generative LLM outputs only the binary judgment without any rationales. {{Question}}, {{Answer 1}}, {{Answer 2}} are the text content of the question, answer 1, and answer 2, respectively.

| Prompt | *As an evaluation expert, given a question and its two possible answers, please choose which answer best aligns with coherence, accuracy, coverage, and overall quality. Output your judgment in JSON format in which "better_answer" is an integer type of 1 or 2, for example, {"better_answer": 1}. Do not include any additional explanations. Below are the question and its candidate answers:* 
 ***Question:*** *{{Question}}* 
 ***Answer 1:*** *{{Answer 1}}* 
 ***Answer 2:*** *{{Answer 2}}* |
| --- | --- |

Table 8: Prompt template for transforming the answers into different formats.

| | |
|---|---|
| Prompt for rewriting the answer into a more verbal format | *You are someone who works on popularizing mathematical knowledge. Please restate the following content in simpler, more accessible language without changing the original meaning, affecting its length, or adding extra information. Below is the input:* {{*Answer*}} |
| Prompt for rewriting the answer into a more formal format | *You are a researcher in the field of mathematics. Please restate the following content using precise mathematical language without changing the original meaning, affecting its length, or adding extra information. Below is the input:* {{*Answer*}} |
| Prompt for rewriting the answer to be more verbose | *Please summarize the input by listing the key points in a numbered format. Below is the input:* {{*Answer*}} |

Table 9: Prompt template for scoring the rationales and judgments of Con-J. {{Judgment}} is the judgment generated by Con-J. the judgment is JSON style with a key named "rationale" that presents Con-J's reasoning and explanation for the task, and another key named "better answer" indicates Con-J's preference prediction.

---

*##Background*
*Given below is a question and two corresponding answers:*
**Question:** *{{Question}}*
**Answer 1:** *{{Answer 1}}*
**Answer 2:** *{{Answer 2}}*
*A judge has assessed these two answers and judged which one is better. Here are its preference prediction (value of "better_answer") and the corresponding rationales (value of "rationale"):*
*{{Judgment}}*
*##Workflow and Scoring*
*Please analyze whether the judgment is correct and evaluate the rationale by scoring them on (i) correctness and (ii) consistency with the binary preference prediction. Here, correctness is defined by whether the rationales accurately understand the content of the question and the answers, provide a reasonable and correct analysis process, and are free of factual errors. Consistency is defined by whether the logical connection between the rationales and the final preference prediction is coherent, and whether both the rationales and the final preference prediction show the same sentiment or attitude towards the two answers.*
*Correctness:*
*1: Completely incorrect.*
*2: Mostly incorrect.*
*3: Partially correct.*
*4: Mostly correct.*
*5: Completely correct.*
*Consistency:*
*1: The sentiment or attitude of the rationales towards either answer (Answer 1 or Answer 2) is inconsistent with the final preference prediction.*
*2: The sentiment or attitude of the rationales towards either answer (Answer 1 or Answer 2) is not clear.*
*3: The sentiment or attitude of the rationales towards either answer (Answer 1 or Answer 2) is consistent with the final preference prediction.*
*##Formatting*
*Please return the results in JSON format, for example: "Preference Prediction Correctness": "Correct", "Rationale Correctness": 4, "Rationale Consistency": 3, where "Preference Prediction Correctness" can be either "Correct" or "Incorrect".*

---

h

Table 10: Judgment accuracy of Con-J and its variants. $*$ indicates the performance difference between Con-J is significant at $p < 0.05$ using a pair-wise t-test.

| Model | Creation | Math | Code |
|---|---|---|---|
| Con-J untrained | $53.6^*$ | $63.4^*$ | $61.7^*$ |
| Con-J w/o Hint | $61.3^*$ | $77.4^*$ | 68.2 |
| Con-J w/o DPO | $54.6^*$ | $64.2^*$ | $63.5^*$ |
| Con-J w/o SFT | 72.2 | $81.3^*$ | 70.3 |
| Con-J w/o rationale | 72.6 | 84.1 | 71.0 |
| Con-J reorder | 72.3 | 85.2 | 70.2 |
| Con-J | 72.4 | 85.0 | 70.1 |

Table 11: Example of rationales and binary preference generated by Con-J. We prompt GPT-4o to annotate the correctness (ranging from 1 to 5) of the rationales and the consistency between the predicted preference and the rationale (ranging from 1 to 3).

| | Prompt input | Rationale | Preferred |
|---|---|---|---|
| **Creation (1)** | **Question:** On the theme of comprehensive rural revitalization, propose a paper title. **Answer 1:** Research on Strategic Pathways for Rural Revitalization from a Multi-Collaborative Perspective. **Answer 2:** Comprehensive Rural Revitalization: Deep Integration of Strategies and Practices. | The second answer provides a broader applicability and depth of description, with a smoother and more natural context than the first answer. ***Correctness: 5, Consistency: 3*** | Answer 2 (correct) |
| **Code (1)** | **Question:** Reverse the elements in an array and make each character in the elements lowercase. planet_arr = ['Earth', 'Saturn', 'Mars']; **Answer 1:** def reverse_array(arr):; return arr[::-1]; def lowercase_array(arr):; return [x.lower() for x in arr]; planet_arr = ['Earth', 'Saturn', 'Mars']; print(reverse_array(planet_arr));print(lowercase_array(planet_arr)); **Answer 2:** planet_arr = ['Earth', 'Saturn', 'Mars']; planet_arr.reverse(); planet_arr = [p.lower() for p in planet_arr]; print(planet_arr); This code will first reverse the order of the elements in the 'planet_arr' array, and then lowercase each character in each element. The resulting array will be ['mars','saturn','earth']. | The first answer provides a clearer step-by-step solution to the problem by achieving array reversal and lowercase conversion through independent functions, which makes the code more modular. The second answer is more concise and accurately meets the question's requirements by first reversing the array and then converting it to lowercase. ***Correctness: 3, Consistency: 1*** | Answer 1 (incorrect) |
| **Math (3)** | **Question:** Today is August 12, 2021. What was the date 30 days ago? **Answer 1:** According to the given information, today is August 12, 2021, so 30 days ago was July 13, 2021. Therefore, 30 days ago was July 13, 2021. **Answer 2:** July 12, 2021. | The second answer does not provide a detailed explanation, whereas the first answer offers a detailed reasoning process and uses correct date subtraction logic. ***Correctness: 2, Consistency: 3*** | Answer 1 (incorrect) |
| **Math (4)** | **Question:** Given the four digits 2, 4, 6, and 7, if a digit can be used more than once in an integer, how many different positive two-digit numbers can be formed? **Answer 1:** . . . This means that each combination (choosing one digit for the first position and another for the second) is treated as a unique number, even if they are mathematically equivalent (like "22" and "44"). There are a total of $C(4, 1) \cdot C(4, 1) = 16$ such combinations (where C denotes combinations). This is the final answer: 16 unique two-digit numbers can be formed. . . . **Answer 2:** . . . If we take order into account (e.g., "24" and "42" are different numbers), we need to use the concept of permutations $P(n, r)$ (the number of ways to arrange r elements from n different elements). In this scenario, we use the formula $P(4, 2) = 4! / (4-2)! = 4 \cdot 3 = 12$ unique arrangements, where "!" denotes factorial. . . . | Answer 1 directly combines the question's requirements to provide the correct solution along with a detailed explanation. ***Correctness: 4, Consistency: 3*** | Answer 1 (correct) |

Table 12: Average performance of Con-J and Con-J w/o rationale on the general test set under the verbosity bias with $\gamma = 1$. The data samples are grouped according to whether the chosen answer is longer than the rejected answer.

| | Chosen Length > Rejected Length | Chosen Length $\leq$ Rejected Length |
|---|---|---|
| Con-J | 0.6795 | 0.5871 |
| Con-J w/o rationale | 0.8301 | 0.4714 |

---

**Algorithm 1** Constructing contrastive judgment pairs for Con-J

---

1: **Input:** $\pi$: a pre-trained LLM; a preference dataset $D = \{(q, a^-, a^+)_i\}_{i=1}^N$.
2: **Output:** $E$: a set of constrastive judgment pairs.
3: $E = \emptyset$
4: **for** $(q, a^-, a^+)_i \in D$ **do**
5:     $p = \text{format}(\text{preamble}, (q, a^-, a^+)_i)$         $\triangleright$ Prompt construction with preamble
6:     Get $M(p)$ with repeated sampling     $\triangleright$ Judgment generation with repeated sampling
7:     $M(p)^+, M(p)^- \leftarrow \text{filter\_correct}(M(p)), \text{filter\_incorrect}(M(p))$   $\triangleright$ Selection with ground truth preference
8:     $E = E \cup \{(j_p, j_n) | j_p \in M(p)^+, j_n \in M(p)^-\}$
9:     $p_p, p_n = \text{format}(\text{preamble\_correct}, (x, a^-, a^+)_i), \text{format}(\text{preamble\_incorrect}, (x, a^-, a^+)_i)$
    $\triangleright$ Using preamble with correct or incorrect hint to construct prompt
10:     Get $M(p_p), M(p_n)$ with hint-driven sampling
11:     $E = E \cup \{(j_p, j_n) | j_p \in M(p_p), j_n \in M(p_n)\}$
12: **end for**
13: **Return** $E$

---

