# OpenReview forum: "Learning LLM-as-a-Judge for Preference Alignment"
_ICLR.cc/2025/Conference — ICLR 2025 Poster_

### Official Review · Reviewer_KgUv · 2024-10-31

**Soundness:** 3
**Presentation:** 3
**Contribution:** 2
**Rating:** 6
**Confidence:** 4

**Summary:**

The paper introduces Con-J, a generative judge trained via self-bootstrapped learning from preference data to overcome limitations of scalar reward models, such as lack of interpretability and dataset bias. This model is trained with a combination of direct preference optimization (DPO) loss for contrastive judgement learning with an additional SFT term over positive judgements. Experiments across various domains (Text Creation, Math, Code) and benchmarks show that Con-J performs effectively, improving rationale correctness during training. This enables accurate judgments and meaningful explanations, supporting human-in-the-loop LLM alignment. Con-J also demonstrates reduced susceptibility to dataset bias compared to scalar and rationale-free models.

**Strengths:**

The paper is clearly-written and the motivation is clear that it attempts to bridge preference modeling with a generative component to reduce bias as well as to improve interpretability. Section 3 is pretty clear as in the math formulation and what the crucial components are in eq.4 and 5.  The empirical evaluation also seems reasonable with a variety of LLMs chosen and data domains being selected. I also find the illustrative examples and prompt-structures to be quite helpful for reproducibility and cross-comparisons.

**Weaknesses:**

I find this work to be a commendable engineering effort but with moderate novelty. In particular, is it not quite evident why do we need DPO’s formulation since the argument about similarity with pretraining objective and DPO’s policy being an optimal solution to the KL-constrained optimization problem is not directly addressed. Furthermore, SFT+ DPO combination for reward model training has been quite popular but I find the motivation for adding the SFT component to be less clear, especially the authors downweight this component with a relatively small weight compared to previous works. The authors argue that adding SFT avoids the reward overoptimization issue but I don't see an empirical evaluation of how this affects the log-likelihoods of pos and neg judgements in their Con-J training.

**Questions:**

1. The paper seeks to address a crucial issue in preference (or reward learning), where the reward model may be prone to bias and is not easily understood from the human interpretability angle. However, the learning algorithm (objective) is not very novel and recent work [1,3-5] has already leveraged such DPO + SFT type losses have leveraged such as contrastive learning formulation. However, I could not find these works being cited unless I am missing something.

2. While the idea is simple, my primary concern is that the generated rationales can be biased or be of low-quality if the chosen base model is not strong enough. Unless I am missing something, the authors do not conduct any human evaluation of the rationales being sampled.  How does one make sure that the rationales are at least surpass some *baseline* reasoning quality, regardless of the preference relation without a manual  verification component?



3. How do you control for the length of the judgements being sampled? From my perspective, if the intention in the Con-J methodology is better/explainable reward learning with compute-efficiency, then would you not have to control for judgement length pre-inference? In that case, for questions that would naturally require a more thorough reasoning process (CoT), wouldn’t the preference judgement be partial or biased in such cases? If judgement-length is controlled, how can shortened or partial/biased judgements’ affect on preference learning be mitigated in light of the inconsistency of rationale vs binary decisions observed in section 4.3?


4 .How was the alpha value chosen? [1] suggests that a larger $\alpha$ (compared to this work) of 0.01 for text-regularization is optimal for generalizable reward learning, including testing on benchmarks like rewardbench. Also, [2,3] provides empirical evidence that DPO loss can lead to reduction in likelihoods of the winning responses. It’s reasonable to expect the SFT component helps in ensuring $\pi_ (j+ / p)$ stays relatively high during optimization, but authors here use an $\alpha$ of $1e^{-6}$ which is significantly lower than the optimal $\alpha$ in [1]. Clearly, this affects SFT-based regularization, but was this $\alpha$ value empirically determined or were there any particular motivations for choosing this value??

---

> ### Author Response · Authors · 2024-11-15
> **A quick question**
>
> Thank you for your valuable comments. I have a quick question: Could you please clarify the references [1-5] you mentioned?
> While I have a general understanding of the main issues you addressed, this information will assist us in preparing our response and refining our paper.

---

> > ### Comment · Reviewer_KgUv · 2024-11-15
> > **Clarification (missing citations from original reviewer comment)**
> >
> > Thank you for pointing out the missing citations. Hope this helps!
> >
> >
> > 1.  Yang, Rui, et al. "Regularizing Hidden States Enables Learning Generalizable Reward Model for LLMs." arXiv preprint arXiv:2406.10216 (2024).
> > 2.  Pal, A., Karkhanis, D., Dooley, S., Roberts, M., Naidu, S., & White, C. (2024). Smaug: Fixing Failure Modes of Preference Optimisation with DPO-Positive. arXiv e-prints, arXiv-2402.
> > 3. Fisch, A., Eisenstein, J., Zayats, V., Agarwal, A., Beirami, A., Nagpal, C., ... & Berant, J. (2024). Robust preference optimization through reward model distillation. arXiv preprint arXiv:2405.19316.
> > 4.  Huang, A., Zhan, W., Xie, T., Lee, J. D., Sun, W., Krishnamurthy, A., & Foster, D. J. (2024). Correcting the mythos of kl-regularization: Direct alignment without overparameterization via chi-squared preference optimization. arXiv preprint arXiv:2407.13399.
> > 5.  Shicong Cen, Jincheng Mei, Katayoon Goshvadi, Hanjun Dai, Tong Yang, Sherry Yang, Dale Schuurmans, Yuejie Chi, and Bo Dai. Value-incentivized preference optimization: A unified approach to online and offline RLHF. arXiv:2405.19320, 2024.

---

> ### Author Response · Authors · 2024-11-22
>
> Thank you for your kind comment and appreciations of our strengths. In the following, we summarize your comment and address the main issues raised.
>
>
> > ***The novelty of Con-J.***
>
> We acknowledge that SFT+DPO is a popular practice while the novelty of Con-J does not lie in this training target. Con-J is a self-bootstrap framework for training a generative judge model based on preference data (framework presented in Figure 2). Unlike traditional methods for training preference judgment models using instruction tunning data, this approach scales better as it is easier to get human's binary preference rather than high-quality verbal feedback.
>
> We also show that Con-J improves its rationale ability with preference learning as a generative judge, potentially facilitating AI transparency and allowing human-AI collaboration in scenarios such as the construction of preference dataset.
>
> > ***The setting of SFT+DPO.***
>
> The general setting of adding an SFT loss term is for stabilizing DPO training by preventing the decrease of the log probability of chosen response (i.e., positive judgment in our paper) [1]. We adopt the SFT+DPO setting as it is a popular practice with both theoretical and empirical evidence. As elaborated in Section A.2, we observe that Con-J shows comparable performance with Con-J w/o SFT in Creation and Code, while outperforms Con-J w/o SFT in Math. Further analysis shows that when trained without SFT, the chosen rewards are relatively small in the Creation and Code datasets, with values of -0.73 and -0.46, respectively, while in the Math dataset, they decrease significantly to -2.37. This implies the benefit brought about by adding SFT term in training on Math dataset.
>
> Here we select the weight of the SFT component according to the reward curve of the chosen response (i.e., positive judgment in our paper). We observed the changes in the chosen reward during training and selected the smallest alpha that would not cause significant changes in chosen reward. This selection results in a relatively small weight than existing work. In previous research of DPO training, the prompts, and chosen responses had diverse formats, whereas in our training, the chosen responses follow a fixed json-style template with a key "rationale" and a key "better_answer". So if the coefficient for the SFT component is too large, it can easily cause the chosen reward to increase rapidly, which could potentially damage the model's performance.
>
> We have revised the relevant discussion in Appendix A.2 and Appendix A.1, but we do not put it in the main text because this training objective is an engineering effort rather than our main contribution. We also added your recommended references and revised the relevant description in Section 3.
>
> [1] Dubey A, Jauhri A, Pandey A, et al. The llama 3 herd of models[J]. arXiv preprint arXiv:2407.21783, 2024.
>
> > ***Human evaluation of the rationales being sampled.***
>
> We add a human evaluation with rationales from Con-J and Con-J untrained when given the same set of 100 randomly sampled  data samples (refer to Section 4.5 and Appendix A.3). Experimental results demonstrated that Con-J has a higher correctness score for rationales compared to Con-J untrained under human evaluation. The performance gap between Con-J and Con-J untrained under human annotations is even greater than that by GPT-4o annotation.
>
>
> > ***The control of judgement length.***
>
> Thanks for the nice idea.
>
> We controlled the output length of Con-J with a fixed length of 512. In our experiments, we observed that when the initial model needs to produce a lengthy judgment, it might get cut off, leading to an "unknown judgment".
> Rendering such judgment as negative judgment is actually beneficial, as when failing to do so, we empirically found that the Con-J's outputs will become increasingly lengthy during preference learning.
>
> > how can shortened or partial/biased judgements' affect on preference learning be mitigated in light of the inconsistency of rationale vs binary decisions?
>
> The model continuously outputs rationale and preference prediction as verbal text, which means the sampled data is coherent. Hence, the model stops generating the rationale only when it thinks a preference prediction should be made.
>
> We admit the judgment filtering process by treating lengthy judgments as unknown and negative may influence the performance on tasks that might need longer outputs to achieve the COT effect. However, our empirical findings show that rationales improve Con-J through a regularization effect during training, but the impact of rationales as a COT effect on Con-J is as significant during inference.
> This is particularly because Con-J serves as a judge, and if the answer parts already include a COT, the judge may not require it as much. To address your concerns, we have  revised the discussions in Appendix A.1 to elaborate on how we control the output length of Con-J and its potential impact.

---

> > ### Author Response · Authors · 2024-11-25
> >
> > Dear reviewer KgUv,
> >
> > We would like to follow up and see whether the response has addressed your concerns. We are grateful for further explanation if you have any addititional questions during the rest of the discussion phase. Thank you once more!

---

### Official Review · Reviewer_sJjd · 2024-11-02

**Soundness:** 3
**Presentation:** 3
**Contribution:** 4
**Rating:** 8
**Confidence:** 4

**Summary:**

The paper introduces Con-J, an approach for training a generative judge using contrastive judgments for preference learning in order to address the problem of (i) lack of interpretability and (ii) susceptibility to biases in scalar reward models. Con-J involves creating contrastive judgments from a set of query-answer pairs sampled from a pre-trained LLM. These contrastive pairs are then used for preference finetuning, for example, direct preference optimization. Finally, they were able to show that Con-J performs better on several benchmarks such as RewardBench.

**Strengths:**

- [S1] The novelty of the paper’s contribution is clear. I appreciate how Con-J’s adjustments of the typical generative RM set-up was motivated.
- [S2] The ablations are also reasonable, especially on the different variants of the contrastive pairs (e.g., positive judgments only, etc.).

**Weaknesses:**

Overall, I lean towards accepting the paper on the merit of its contribution's novelty (new way of framing the generative RM setup), but there are some sections--particularly in the analysis--that are not fully convincing and needs revisiting (see [W1]).

[W1] I’m not entirely convinced with the analyses of Consistency and Correctness in Sections 4.3 and 4.5. I think it’s helpful if the paper first defines what “consistency” and “correctness” measure. In addition, it will be helpful to show that the GPT-4o scorer is reliable. Suggestion to the authors: define and motivate the aspects of consistency and correctness in the beginning of the paper.
- The prompt template in Table 9 speaks about the emotional tone, which I think measures a different aspect than consistency. Perhaps that’s also the reason why the results  in Figure 3b are surprising (I would expect the consistency of the rationales would also improve) and aren’t consistent with the cited literature on chain-of-thought (CoT).
- The scoring for Correctness is also confusing: the rubric for 3 can be confused for 2 and 4.  It’s also important to describe what correctness means for a subjective task such as Creation. I wonder what the findings would be if the rubric is more distinct.

**Questions:**

Questions
- [Q1] Can you clarify (or point to a URL) how you were able to obtain a test split for Ultrafeedback? The HuggingFace version (openbmb/Ultrafeedback) only contains a train split.

Comments/Nits:
- [C1] Line 259-261. The “Code” vertical domain is not mentioned
- [C2] Some LaTeX citation commands were used incorrectly. For example, \citet is often used in place of \citep, breaking the flow of writing. I highly recommend the authors to revisit these parts and update accordingly.

---

> ### Author Response · Authors · 2024-11-22
>
> Thanks for your kind comments. In the following, we address the main issues raised.
>
> > I’m not entirely convinced with the analyses of Consistency and Correctness in Sections 4.3 and 4.5. I think it’s helpful if the paper first defines what “consistency” and “correctness” measure. Suggestion to the authors: define and motivate the aspects of consistency and correctness in the beginning of the paper.
>
> We add the definitions of consistency and correctness at the beginning of the paper. Briefly, correctness refers to whether the rationale provides an accurate analysis supporting the preference prediction, while consistency refers to whether the rationale expresses the same preference as Con-J's final preference prediction.
>
> > In addition, it will be helpful to show that the GPT-4o scorer is reliable.
>
> We have added an additional human evaluation to collect human annotations of consistency and correctness (see Appendix B.3). We observe that the Krippendorff's $\alpha$ values between the GPT-4o annotators and human annotators are 0.4427 for correctness and 0.6495 for consistency, indicating moderate and substantial agreement, respectively. This suggests a consistent trend between human annotators and GPT-4o, although some disagreements remain.
>
> > The prompt template in Table 9 speaks about the emotional tone, which I think measures a different aspect than consistency. Perhaps that’s also the reason why the results in Figure 3b are surprising (I would expect the consistency of the rationales would also improve) and aren’t consistent with the cited literature on chain-of-thought (CoT). The scoring for Correctness is also confusing: the rubric for 3 can be confused for 2 and 4. I wonder what the findings would be if the rubric is more distinct.
>
> Thanks for pointing out the issue in our prompt template, we acknowledge that the design of prompt can affect the performance. Therefore, we have refined the prompt template in Table 9 with detailed definitions of correctness and consistency as well as descriptions of different consistency levels instead of using "emotional tone".
>
> With the new prompts, similar experimental results are observed. There still exists minor difference, such as a general increase in correctness and a drop in consistency for all checkpoints with the new templates. Although these minor differences still exist, the overall experimental observations remain similar. Specifically, correctness improves while consistency slightly declines during preference learning of Con-J.
>
> We have added a qualitative analysis to investigate the drop in consistency during preference learning and illustrate several solutions to mitigate this issue (see Section A.3.2 and Section A.4). The definition of consistency is whether the rationale and the final preference prediction have the same attitude/preference towards either answer. We observe that when inconsistency happens, the performance of Con-J also drops, which indicates that Con-J refuses to express its attitude towards either answer when it lacks the ability to make judgments. This indicates that the low consistency may not be bad, but potentially indicates that the model is not convincing in its judgment on certain cases and calls for more human interference. In future work, a more effective data construction process may address this problem by designing more appropriate expected rationales for these situations, such as having the model output "I find it difficult to judge, but I can provide some analysis ..."".
>
> > It’s also important to describe what correctness means for a subjective task such as Creation.
>
> We have currently conducted experiments only in Math dataset because it allows for an objective analysis on the quality of rationales in the context of math problem-solving. In contrast, the Creation dataset is more subjective. How to evaluate rationales in a potentially subjective scenario is left for future work.
>
> > Can you clarify (or point to a URL) how you were able to obtain a test split for Ultrafeedback?
>
> We split Ultrafeedback according to https://huggingface.co/datasets/HuggingFaceH4/ultrafeedback_binarized. We have attached the link to our paper.
>
> > Line 259-261. The “Code” vertical domain is not mentioned.
>
> We have mentioned the Code dataset at Section 4.1. We have polished the paragraph for a clearer description of the dataset.
>
> > Some LaTeX citation commands were used incorrectly
>
> We have gone through the paper and carefully revised the usage of LaTeX citations.

---

> > ### Author Response · Authors · 2024-11-25
> >
> > Dear reviewer sJjd,
> >
> > We would like to follow up and see whether the response has addressed your concerns. We are grateful for further explanation if you have any addititional questions during the rest of the discussion phase. Thank you once more!

---

> > ### Comment · Reviewer_sJjd · 2024-12-02
> >
> > Thank you for acknowledging the review and updating the manuscript. I have increased my score as I believe that this work is directionally a positive improvement in the field.

---

### Official Review · Reviewer_21Wd · 2024-11-03

**Soundness:** 2
**Presentation:** 3
**Contribution:** 3
**Rating:** 6
**Confidence:** 3

**Summary:**

This paper introduces Con-J, a new way of learning preferences by augmenting human pairwise preferences with LM-generated rationales. First, a LM is prompted to generate rationale pairs ($j^+$, $j^-$) for either element of the preference pair ($a^+$, $a^-$). The base LM is then trained with DPO to optimize $j^+$ over $j^-$ (vs. DPO with scalar rewards, which is simply trained to optimize $a^+$ over $a^-$). Empirical studies are performed on three domains (Creation, Math, Code), which find that Con-J outperforms scalar reward models at providing the correct judgment, that it outperforms other generative judges (Auto-J and Prometheus 2), and and that it is less susceptible to bias and spurious correlations in the reward function.

**Strengths:**

The experimental results demonstrate that Con-J is an effective method, compared to scalar reward models. The results section is quite comprehensive, with baselines across multiple models and datasets. The analysis section is informative with lots of useful takeaways: including studies of how Con-J can be robust in the presence of bias, and where LMs can generate rationales that fail to support the answer judgment. Overall, the claims are well-supported.

**Weaknesses:**

1. The empirical results (Table 10, Figure 4(b)(d)) seem to indicate that the main contribution to Con-J's good results are not the rationales, but (presumably) the regularization in eq. 9 instead (It's unclear to me what exact parts Con-J without rational removes...). Experimentally, what happens when this regularization is ablated? Furthermore, the claims in the abstract/introduction seem to suggest that rationales themselves are the reason why Con-J succeeds. While experimentally it's demonstrated that the rationales do improve robustness to bias, it seems misleading to claim this when the main results actually suggest that the rationales do not improve judgment accuracy at all. It may be good to do a deeper analysis of what is actually inducing this improvement.
2. Furthermore, the introduction highlights interpretability as a benefit of rationales (and puts it on the same priority level as robustness to bias). While intuitively rationales are indeed more interpretable, the paper never demonstrates the utility of this interpretability. What happens if you do have humans (or even other LMs) check the rationales and throw out inconsistent ones? (As far as I understand, the filtering step simply checks the correctness of the final answer, but not the rationale itself.) Is this interpretability actually useful for constructing better / more useful reward models? Will it be better to align downstream models with rationales?
3. The method has many moving parts. While the figure is useful, parts of the text and figure could be clearer. Having a concrete example with a LM-generated rationale in the main body or the figure could be useful for visualizing the method. Furthermore, it may be good to list out all the moving parts, esp for the ablations it is unclear what parts are preserved and what parts are not (e.g. how exactly does Con-J without rationales differ from the scalar model).
4. Note -- quite a few generative reward model methods have come out at about the same time this work (see https://arxiv.org/abs/2408.15240, https://arxiv.org/abs/2410.12832). While these can be considered concurrent work, perhaps a discussion of the tradeoffs of this method compared to other methods could be useful somewhere in the paper.

**Questions:**

1. How exactly does Con-J without rationale differ from the scalar model (Figure 4)?
2. One can also imagine a debiasing effect from using pre-trained LMs as rationale generators -- they are likely to generate rationales that align with preferences from their pre-trained data. While this may be ok if the preferences are generally in line with the pre-trained data, there may be issues with aligning to preferences with fundamentally different values or principles. Is there any investigation of whether the LM likes to generate the same set of values in its rationales?
3. Are there overall statistics from section 4.5 about how consistent / correct the rationales tend to be? Either automated, or human annotated (perhaps on a subset).

---

> ### Author Response · Authors · 2024-11-22
>
> Thank you for your kind comment and appreciation of our strengths. In the following, we address the main issues raised (some of your questions are re-arranged in order to facilitate a progressive reading of our responses).
>
> > How exactly does Con-J without rationale differ from the scalar model (Figure 4)?
>
> Con-J without rationale is trained and evaluated with the prompt template presented in Table 7. This indicates that Con-J is prompted to output its preference prediction directly without generating any rationales.
>
> Con-J still differs from the scalar model in that it still leverages the generation capability of the original LLM, while the scalar model replaces the LM head with a value head. Therefore, Con-J w/o rationale lacks the impact of the regularization effect on generating the rationales during training, while SM lacks the regularization effect due to the initialization of the value head. This difference results in a variation in their performance.
>
> >The empirical results (Table 10, Figure 4(b)(d)) seem to indicate that the main contribution to Con-J's good results are not the rationales, but (presumably) the regularization in eq. 9 instead (It's unclear to me what exact parts Con-J without rational removes...). Experimentally, what happens when this regularization is ablated? Furthermore, the claims in the abstract/introduction seem to suggest that rationales themselves are the reason why Con-J succeeds. While experimentally it's demonstrated that the rationales do improve robustness to bias, it seems misleading to claim this when the main results actually suggest that the rationales do not improve judgment accuracy at all. It may be good to do a deeper analysis of what is actually inducing this improvement.
>
> To investigate this problem, we conducted an analysis on cases in the general test set where Con-J outperforms Con-J w/o rationale under the verbosity bias with $\gamma=1.0$. Interestingly, amongst these cases, 79.2% of the chosen answers are shorter than the rejected ones. In contrast, in cases where Con-J underperforms Con-J w/o rationale, only 31.6% of the chosen a are shorter than the rejected ones.
>
> We further present the table below showing the accuracy  of Con-J and Con-J w/o rationale in cases where the chosen answer is shorter/longer than the rejected one. While the two settings have close overall performance, they behave differently under the two scenarios. Con-J w/o rationale performs well when the correct answer is longer than the rejected one but notably worse when the rejected answer is longer - a manifestation of verbosity bias. A similar trend is observed for Con-J under the two cases, but with a far less remarkable gap, indicating a strong bias counteracting effect of the generated rationale. This finding suggests that even when there is no difference in the average performance of Con-J and Con-J w/o rationale, the influence of verbosity bias on them could be different. We have revised our paper by adding this observation.
>
> |                     | Chosen Length > Rejected Length | Chosen Length <= Rejected Length |
> | :------------------ | :-----------------------------: | -------------------------------: |
> | Con-J               |             0.6795              |                           0.5871 |
> | Con-J w/o rationale |             0.8301              |                           0.4714 |
>
> >What happens if you do have humans (or even other LMs) check the rationales and throw out inconsistent ones? (As far as I understand, the filtering step simply checks the correctness of the final answer, but not the rationale itself.) Is this interpretability actually useful for constructing better / more useful reward models? Will it be better to align downstream models with rationales?
>
> To address this issue, we have made efforts in two aspects.
>
> First, we have conducted human evaluations on the correctness and consistency of these rationales and provided some qualitative analysis (detailed in Section 4.3 and Section A.3). The human evaluation has provided some valuable insights. For example, when the rationale highlights some issues in the answers, humans can quickly verify whether they truly exist and assess the judgment accuracy; conversely, when the rationale refuses to explicitly express its attitude towards either answer, it is likely that Con-J lacks sufficient capability to make a correct judgment. This facilitates the construction of preference data for RLHF training via human-machine collaboration, which makes the annotation process more transparent. Similar approach has been applied to industrial pipelines such as [1].
>
> Second, we conducted an experiment to investigate whether rationales with controlled quality can enhance Con-J's ability (see Section 4.3 and Appendix A.4). Generally, we observe that Con-J trained with rationales filtered by their correctness and consistency scores is also better at making preference judgments.

---

> > ### Author Response · Authors · 2024-11-22
> >
> > > The method has many moving parts. While the figure is useful, parts of the text and figure could be clearer. Having a concrete example with a LM-generated rationale in the main body or the figure could be useful for visualizing the method. Furthermore, it may be good to list out all the moving parts, esp for the ablations it is unclear what parts are preserved and what parts are not (e.g. how exactly does Con-J without rationales differ from the scalar model).
> >
> > A concrete example is presented in Figure 1, so we abstract Figure 2 (the method figure) away from real examples. To better visualize the method, we refine Figure 2 by deducing some moving parts and splitting the judgment content into a preference prediction part and a rationale part.
> >
> >
> > > Note -- quite a few generative reward model methods have come out at about the same time this work (see https://arxiv.org/abs/2408.15240, https://arxiv.org/abs/2410.12832). While these can be considered concurrent work, perhaps a discussion of the tradeoffs of this method compared to other methods could be useful somewhere in the paper.
> >
> > Thank you for pointing out the recent related work. A significant difference between our work and previous research like https://arxiv.org/abs/2408.15240 is that we employ a contrastive learning strategy to learn from preference data, while this approach utilizes both self-generated positive and negative judgments.
> >
> > As a concurrent work, https://arxiv.org/abs/2410.12832 utilizes a similar self-bootstrapping idea as Con-J with some differences in the implementation of constructing contrastive judgments. One of the key difference lies in that we conducted more detailed analyses on the impact of bias and conducted evaluation with GPT-4o and humans to study the quality of the rationales. Additionally, we showed that Con-J's ability to generate rationales and preference prediction can co-evolve in preference learning, while controlling the quality of the self-bootstrapped rationales can further enhance the effectiveness of preference learning.
> >
> > We have revised our paper to include the comparison with these works in Section 1 and Section 5.
> >
> > > One can also imagine a debiasing effect from using pre-trained LMs as rationale generators -- they are likely to generate rationales that align with preferences from their pre-trained data. While this may be ok if the preferences are generally in line with the pre-trained data, there may be issues with aligning to preferences with fundamentally different values or principles. Is there any investigation of whether the LM likes to generate the same set of values in its rationales?
> >
> > We tested this issue in the context of format bias (adversarial), with $\gamma$ = 0.33. Experimental results showed that, in cases where Con-J outperforms SM, the untrained model achieves an accuracy of 85.6%. In contrast, in cases where Con-J underperforms SM, the untrained model has an accuracy of 49.8%. This suggests that Con-J is likely to generate rationales aligning with preferences from its pre-trained data.
> >
> > However, this does not mean that Con-J's ability to resist bias compared to SM is entirely due to the initial model. As shown in Figure 4(b), Con-J's performance is noticeably better than the initial model when $\gamma <= 0.67$, indicating Con-J also improves from the biased data. Moreover, even in adversarial scenarios, there are cases (1-85.6%=14.4%) where Con-J outperforms both the initial model and SM.
> >
> > Finally, we agree that it might be challenging to align LLMs to preferences with fundamentally different values or principles. We have revised the paper by adding relevant discussion in Section 4.4.
> >
> > > Are there overall statistics from section 4.5 about how consistent / correct the rationales tend to be? Either automated, or human annotated (perhaps on a subset).
> >
> > The average correctness score and consistency score are presented in Figure 3(a) and Figure 3(b) (the last checkpoint). Notes have been added in Section 4.3. The average correctness score and consistency score are 3.875 and 2.875 in a subset with 100 data samples, respectively (see Appendix A.3).
> >
> > [1] McAleese N, Pokorny R M, Uribe J F C, et al. Llm critics help catch llm bugs[J]. arXiv preprint arXiv:2407.00215, 2024.

---

> > > ### Author Response · Authors · 2024-11-25
> > >
> > > Dear reviewer 21Wd,
> > >
> > > We would like to follow up and see whether the response has addressed your concerns. We are grateful for further explanation if you have any addititional questions during the rest of the discussion phase. Thank you once more!

---

### Official Review · Reviewer_eNC9 · 2024-11-07

**Soundness:** 3
**Presentation:** 3
**Contribution:** 3
**Rating:** 6
**Confidence:** 4

**Summary:**

The paper proposes a generative approach to build reward models for preference learning. They generate judgements with explanations for both positive and negative samples in preference pairs and then use DPO to align a generative reward model. They find that this contrastive approach leads to improved performance on internal and public benchmarks when compared against scalar reward models and other generative methods.

**Strengths:**

* Very interesting and relevant approach that uses synthetic data to effectively propose a good empirical solution.
* Generating rationales would be very useful to verbalize and detect biases in the preference data and can help provide better data quality.
* I like that the authors train a model on publicly available datasets and report numbers on widely used benchmarks instead of just relying on the internal test sets.

**Weaknesses:**

* The approach feels like it would add a lot of overhead in alignment training - RLHF is already extremely computationally intensive and waiting for models to generate rationales before providing the judgement feels like it would not be a feasible approach to use for cutting-edge model alignment.
* I would like to see some qualitative analysis of the rationales - there seems to be minimal information on how good they are and what they might be useful for. Has there been any human evaluation of the rationales when inconsistencies have been detected?

**Questions:**

Please see weakness section.

---

> ### Author Response · Authors · 2024-11-22
>
> Thanks for your kind comment and appreciation of our approach and experiments on open-source datasets. In the following, we address the main issues raised.
>
> > The approach feels like it would add a lot of overhead in alignment training - RLHF is already extremely computationally intensive and waiting for models to generate rationales before providing the judgement feels like it would not be a feasible approach to use for cutting-edge model alignment.
>
> The practice of RLHF actually involves two important steps: constructing preference data, and training the policy model based on its basis. We discuss the feasibility of Con-J in both steps below.
>
> First, the primary goal of Con-J is to address the challenge of constructing and scaling preference data as a machine annotator. In industry, the construction of preference data typically combines both human annotations, which have a higher alignment with human value, and machine annotations which could be more scalable. While a scalar model does not provide explanations on its annotations, Con-J makes it happen by providing machine-generated rationales on human preferences, contributing to a more transparent annotation process. A similar approach has been applied to industrial pipelines such as annotating code data [1].
>
> Second, we admit that Con-J is less effective than the scalar reward model in RLHF training. However, the majority of time is spent on the policy model's inference process rather than on the reward model during RLHF. To address the efficiency issue, Con-J can be combined with some mechanism that could determine whether generating the rationales preceding the preference prediction is necessary. Since the RLHF training process is beyond the scope of this paper, we leave the combination of Con-J and the RLHF process as future work.
>
> [1] McAleese N, Pokorny R M, Uribe J F C, et al. Llm critics help catch llm bugs[J]. arXiv preprint arXiv:2407.00215, 2024.
>
> > I would like to see some qualitative analysis of the rationales - there seems to be minimal information on how good they are and what they might be useful for. Has there been any human evaluation of the rationales when inconsistencies have been detected?
>
> We performed a human evaluation to investigate the correctness and consistency of the rationales. We recruited six annotators from college students to evaluate 100 randomly sampled rationales generated by Con-J in terms of their correctness and consistency with the preference prediction.
>
> As a result, the average consistency value is above 2.85, indicating a high consistency level. We further conducted a qualitative analysis on the very few cases with consistency scores of 2 or 1. We find that in these cases, Con-J often chooses to refrain from clearly expressing its preference, e.g., refuses to indicate which answer to refer to: 'The answer provided a clear calculation process and gave an accurate answer, with more standard mathematical expressions and a clear logical order.', and presents the weakness of both answers: "Answer 1 did not accurately understand the original mathematical problem; Answer 2 overly complicated the solution form, neglecting a more straightforward calculation method."
>
> Furthermore, we found that the consistency of human-annotated cases has a moderate agreement with GPT-4's annotations (Cohen's Kappa=0.47). Therefore, we extended this analysis to examine cases in the full test set that GPT-4 annotated as having low consistency. We found that in cases with low consistency, the accuracy of the preference prediction is 59.9%, which is significantly lower than on the whole test set (85.0%). This suggests that low consistency generally occurs when Con-J lacks sufficient judgment ability, resulting in the generated rationale lacking an explicit attitude towards either answer and supporting the final preference prediction.
>
> We have revised our paper and presented the detailed design of the human evaluation and the qualitative analysis regarding the rationale (including examples) in Section 4.5 and Appendix A.3.

---

> > ### Author Response · Authors · 2024-11-25
> >
> > Dear reviewer eNC9,
> >
> > We would like to follow up and see whether the response has addressed your concerns. We are grateful for further explanation if you have any addititional questions during the rest of the discussion phase. Thank you once more!

---

### Author Response · Authors · 2024-11-22

We thank all reviewers for their constructive comments and consistent appreciation of our strengths such as "clear novelty", "interesting/relevant/effective methods" and "well-supported experiment". We present the major changes made to the revision which are marked in Blue. We have also responded to each of your concerns under your respective reviews. The main changes are listed as follows:

- We have added a human evaluation in Section 4.5 and Section A.3 to investigate the correctness and consistency of the rationales. We recruited 6 annotators from college students to annotate 100 randomly selected data samples with rationales generated by Con-J and Con-J untrained. Experimental results suggest that annotations from GPT-4o have the potential to substitute for human annotators, despite some disagreements, for which we provide a more detailed analysis.
- We conducted a quantitative analysis in Section 4.5 and Section A.3 with 3 out of the 6 annotators discussing cases where rationales are (1) useful and high-quality, (2) useless with low correctness or consistency, and (3) show disagreement between human annotators or between humans and GPT-4o.
- We added an experiment in Section 4.3 and Section A.4 with a subset of 1,2000 data samples randomly selected from the Math dataset, showing that controlling the correctness of the rationales during preference learning can further enhance the effectiveness of preference prediction with the same scale of training data.
- We added analysis regarding the bias effect, refined Figure 2, refined the prompts for GPT-4o scoring, and fixed minor issues such as citations.

---

### Meta-Review · Area_Chair_7311 · 2024-12-18

**Metareview:**

This paper introduces Con-J, a method for aligning LLMs with human values that uses a generative LLM to produce contrastive judgments with natural language rationales, addressing the limitations of traditional scalar reward models. Training this "Judge" with Direct Preference Optimization results in improved interpretability and robustness against biases, outperforming scalar models and other generative LLMs in experiments. The authors release their code and trained models. Most reviewers appreciate the novelty of the approach (one reviewer disagrees), the clarity of presentation, as well as the evaluation and results. However, the reviewers also raise some concerns, primarily regarding the rationales and interpretability (the analysis and conclusions is not convincing) and the analysis of consistency and correctness. The authors respond to all concerns and questions and one reviewer increases their score; not all reviewers engaged in discussion.

**Additional Comments On Reviewer Discussion:**

The discussions mostly focus around the weaknesses mentioned by the reviewers. The authors respond to each of them in detail, conducting additional experiments when needed. Overall the author responses seem to adequately address the weaknesses and questions of the reviewers.

---

### Decision · Program_Chairs · 2025-01-22

Accept (Poster)